# Characterizing and Mapping Volcanic Flow Deposits on Mount St. Helens via Dual-Band SAR Imagery

Nikola Rogic [1,*], Sylvain J. Charbonnier [1], Franco Garin [1], Guy W. Dayhoff II [2], Eric Gagliano [3], Mel Rodgers [1], Charles B. Connor [1], Sameer Varma [4] and David Shean [3]

1 School of Geosciences, University of South Florida, Tampa, FL 33620, USA; sylvain@usf.edu (S.J.C.); francoalainv@usf.edu (F.G.); melrodgers@usf.edu (M.R.); cbconnor@usf.edu (C.B.C.)
2 Department of Chemistry, University of South Florida, Tampa, FL 33620, USA; gdayhoff@usf.edu
3 Department of Civil and Environmental Engineering, University of Washington, Seattle, WA 98195, USA; egagli@uw.edu (E.G.); dshean@uw.edu (D.S.)
4 Department of Molecular Biosciences, University of South Florida, Tampa, FL 33620, USA; svarma@usf.edu
* Correspondence: nrogic@usf.edu; Tel.: +1-(813)-593-0800

**Abstract:** Mapping volcanic flow deposits can be achieved by considering backscattering characteristics as a metric of surface roughness. In this study, we developed an approach to extract a measure of surface roughness from dual-band airborne Synthetic Aperture Radar (ASAR) backscattering data to characterize and map various volcanic flow deposits—namely, debris avalanches, lahars, lava flows, and pyroclastic density currents. We employed ASAR and Indian Space Research Organization (ISRO) airborne SAR datasets, from a joint project (ASAR-ISRO), acquired in December 2019 at 2 m spatial resolution, to assess the role and importance of incorporating dual-band data, i.e., L-band and S-band, into surface roughness models. Additionally, we derived and analyzed surface roughness from a digital surface model (DSM) generated from unoccupied aircraft systems (UAS) acquisitions using Structure from Motion (SfM) photogrammetry techniques. These UAS-derived surface roughness outputs served as meter-scale calibration products to validate the radar roughness data over targeted areas. Herein, we applied our method to a region in the United States over the Mount St. Helens volcano in the Cascade Range of Washington state. Our results showed that dual-band systems can be utilized to characterize different types of volcanic deposits and range of terrain roughness. Importantly, we found that a combination of radar wavelengths (i.e., 9 and 24 cm), in tandem with high-spatial-resolution backscatter measurements, yields improved surface roughness maps, compared to single-band, satellite-based approaches at coarser resolution. The L-band (24 cm) can effectively differentiate small, medium, and large-scale structures, namely, blocks/boulders from fine-grained lahar deposits and hummocks from debris avalanche deposits. Additionally, variation in the roughness estimates of lahar and debris avalanche deposits can be identified and quantified individually. In contrast, the S-band (9 cm) can distinguish different soil moisture conditions across variable terrain; for example, identify wet active channels. In principle, this dual-band approach can also be employed with time series of various other SAR data of higher coherence (such as satellite SAR), using different wavelengths and polarizations, encompassing a wider range of surface roughness, and ultimately enabling additional applications at other volcanoes worldwide and even beyond volcanology.

**Keywords:** remote sensing; dual band; radar backscatter; surface roughness; volcanic deposits mapping; UAS

## 1. Introduction

Remote sensing is an essential tool for studying active subaerial volcanoes [1,2]. Passive and active remote sensing imagery (i.e., optical, thermal, and radar) can be used to map structural features and eruptive volcanic deposits [3]. This imagery is especially

useful in areas where ground-based monitoring and field campaigns would be infeasible or hazardous. While optical sensors are important tools, their usage can be hindered by cloud cover. Synthetic Aperture Radar (SAR) instruments, in contrast, can acquire information independent of meteorological or illumination conditions, making them a powerful alternative. Indeed, Interferometric SAR (InSAR) data, obtained using monostatic or bistatic acquisitions, has become an invaluable asset for remote terrain mapping, as it can be used to produce digital elevation models and track topographic changes [4,5]. Nonetheless, most studies employing SAR instruments rely on single wavelength (e.g., C-, X-, or L-band) SAR data to identify and map topographic features and changes.

In this study, we used National Aeronautics and Space Administration (NASA) airborne SAR (ASAR) and Indian Space Research Organization (ISRO) datasets (Version 1.3) [6] to generate dual-band surface roughness maps and assessed their ability to differentiate volcanic terrains—debris avalanches, lahars, lava flows, and pyroclastic density currents. Dual-frequency airborne SAR systems, like those used to collect the ASAR-ISRO data presented here, use two different wavelengths in contemporaneous data acquisition with 2 m spatial resolution, which can be used to discriminate surface roughness, relative to each radar wavelength and based on different backscattering characteristics, without temporal differences hampering interpretation. We refer to surface roughness as irregularities or variations detected by radar sensors in terms of surface height or slope, where factors such as the surficial mutability of rocks, vegetation, or water bodies within themselves or in comparison to one another can contribute to these variations, which are quantified, visualized, and assessed following the methodology described in this study. In radar-based remote sensing, surface roughness is often quantified using the radar backscatter coefficient, which measures the strength (amplitude) of the radar signal reflected back to the sensor. Smooth surfaces typically reflect less radar energy (lower amplitude) back to the sensor compared to rough surfaces (higher amplitude), as the radar waves scatter in multiple directions [7–9]. Consequently, a higher backscatter coefficient usually indicates a rougher surface. It is important to note that the wavelength of the radar signal also plays a role in the interaction with the surface, as it determines the scale of surface features that the radar can detect. Different radar systems operate at various wavelengths, and choosing the appropriate wavelength is essential for effectively studying specific surface features. Therefore, both the amplitude and wavelength of the ASAR-specific signal are prime factors in the characterization of surface roughness in our study. Mapping and identifying volcanic mass flow deposits during volcanic events is essential to better understand the extent, timing, and magnitude of topographic changes they can generate. Many recent studies used backscatter and/or a measure of surface roughness to examine lava flows [10–15], lahars [4,16,17], and pyroclastic density currents [4,18–20], among other features [21], by means of a single SAR image examination, composite image, RGB color maps, or ratio maps, respectively. Some of these studies used both amplitude and phase information [4] in their characterization of surface textures and materials. Additionally, it has been argued that a combination of optical and SAR products can have beneficial impact on volcanic flow mapping [22], which can be applied in the areas recently impacted by volcanic hazards, as outlined by recent studies using such approaches for rapid volcanic hazard assessment [23]. Unlike ASAR instruments, which are deployed onboard aircraft at high altitudes, unoccupied aircraft systems (UAS) can be deployed at low altitudes to yield centimeter-scale digital surface models (DSMs) using Structure from Motion (SfM) photogrammetry [24]. Furthermore, real-time kinematic (RTK) enabled UAS are capable of robust georeferencing, where their high geospatial positioning accuracy is founded upon low root-mean-squared error measurements in the range of millimeters to centimeters (in the single digits) [25]. DSMs produced by RTK-enabled UAS could potentially act as substructures in the generation of ground-truth roughness models that can be used to calibrate ASAR-ISRO data.

The types of radar backscatter [26] assessed in this study and linked to a particular feature related to (i) specular scattering, such as from the flat surface of a lake (low backscatter, dark areas in SAR image), (ii) volumetric scattering, found in regions of tall vegetation,

for example, (iii) surface scattering that may relate to variable surfaces of short vegetation, (iv) single-bounce scattering for geometric features such as mountain faces lacking vegetation, and (v) double-bounce scattering in areas with flat, perpendicular reflectors, such as cities and urban areas. Different backscattering properties and characteristics are used to distinguish between targets of interest, such as vegetated terrain (high backscatter) and bare earth terrains (low backscatter), for example. For radar images of non-vegetated soils, the amplitude of the radar signal return depends on several important factors, including (i) surface roughness, (ii) soil moisture, (iii) incidence angle, (iv) radar wavelength, and (v) polarization of the radar signal [19,27,28].

It has been argued that cross-polarized data, such as horizontal transmit and vertical receive (HV) or vertical transmit and horizontal receive (VH) amplitude imagery, are more likely to show differences between vegetated and barren ground than the single-polarized data (i.e., HH and/or VV amplitude imagery) [22]. In addition, over dry areas where moisture content does not vary significantly, radar backscatter is primarily determined by Bragg resonance [8,29]. If all other conditions are the same, surfaces consisting of many scattering elements with sizes close to the radar wavelength are expected to have high backscatter. Challenges that could influence interpretation of airborne SAR data may include seasonal snow and/or ice cover or steep slopes near the volcano summit, introducing radar shadows [22], especially in shorter wavelength (e.g., S-band in this study), which could add complexity to data processing. Moreover, there may be issues involving incidence angles and terrain aspect, position and orientation uncertainty, turbulence introducing 'jitter', and other artifacts in the SAR images.

Surface roughness in volcanic mass flow deposits has been investigated before [30–35], utilizing surface roughness in volcanic flow mapping, which can facilitate quantitative methods of deposits differentiation [8,36,37]. Key areas of volcano science that may benefit significantly from utilization of high-resolution dual-band SAR data used here are not only those characterizing and mapping volcanic flow deposits through SAR backscattering properties but also via the differentiation of their key features. By analyzing deposits with a wide range of grain sizes (sub-millimeters to meters), sorting, and thicknesses (centimeters to meters), we aim to build a database of scattering characteristics (based on calibrated radar backscatter coefficients for each wavelength) for volcanic flow deposits including lahar (water-saturated mass flow) deposits, lava flows, debris avalanches, and pyroclastic density current (PDC, gas-solid mass flow) deposits. Such a database will provide a physical basis for interpretation of the radar scattering mechanisms mentioned above (i.e., single vs. double bounce, volume vs. surface).

In this work, we first elaborated a method to compute a measure of surface roughness from dual-band ASAR-ISRO data. To calibrate our roughness outputs, we then generated a UAS-based DSM produced through Structure from Motion (SfM) photogrammetry, which covers a portion of distinct hummocky terrain within the study site, thus, serving as a geomorphologically appropriate testable area due to its relatively consistent patterns of hypsometric variability. From these DSMs, we constructed ground-truth roughness models that were then qualitatively and quantitatively evaluated to complete our calibration approach. We show the main ASAR-ISRO roughness results obtained over two areas of Mount St. Helens volcano (USA), using both polarimetric decompositions of the backscatter signal and roughness estimates obtained with our protocol. Our results provide important constraints on the identification and differentiation of various volcanic deposits used for mapping, statistical, and modeling applications.

## 2. Materials and Methods

### 2.1. Airborne SAR Data Properties

In this study, we used dual-band (L + S) ASAR-ISRO [6,38], a precursor mission to the spaceborne dual-frequency (L + S) NASA ISRO Synthetic Aperture Radar (NISAR). The dual-frequency (L + S) ASAR operated at an altitude of 8 km with a platform velocity of 120 m s$^{-1}$. The center frequencies were at 3200 MHz (S-band SAR) and 1250 MHz

(L-band SAR), and in addition to a simultaneous mode of operation, the system could also be operated in 'L-band only' and 'S-band only' modes. The ASAR specifications are comparable to those of the future NISAR mission expected to launch in 2024. We used the (L + S) ASAR-ISRO data to prepare a single product, highlighting features and characteristics in both bands. Specifically, the type of (L + S) ASAR-ISRO data used here is a Level 2 geocoded product, providing multi-look amplitude images in geocoded coordinates. Such data, along with Equation (2), were delivered as a package product, both terrain- and geometrically corrected by the NASA-ISRO [6,38], where nominal map projections available for the image products are in UTM. The images were resampled (i.e., cubic convolution) to support spacing requirements in range and azimuth direction and were provided in a GeoTIFF format. The system can operate in L-band or S-band only, as well as in a simultaneous mode operation (L + S). The (L + S) ASAR polarization modes available are single (HH/VV), dual (HH + VV/VV + VH), compact (RH + RV/LH + LV); quasi-quad pol, and full pol (HH + HV + VH + VV). We used full-pol datasets in our analyses. The incidence angle for each pixel was provided alongside with the (L + S) ASAR-ISRO product in a GeoTIFF file, and the value of incidence angles for our datasets fell within a range between 24° and 77°.

The target area in our study was a region over Mount St. Helens in the Cascade Range of Washington state (Figure 1). Mount St. Helens, a 2549 m (asl) high stratovolcano located in the Cascades Volcanic Arc, was a perfect candidate, as it is the most active volcano in the continental United States. It produced a Volcanic Explosivity Index (VEI) 4 eruption in 18 May 1980, which is considered the deadliest and most economically destructive volcanic event in U.S. history [39].

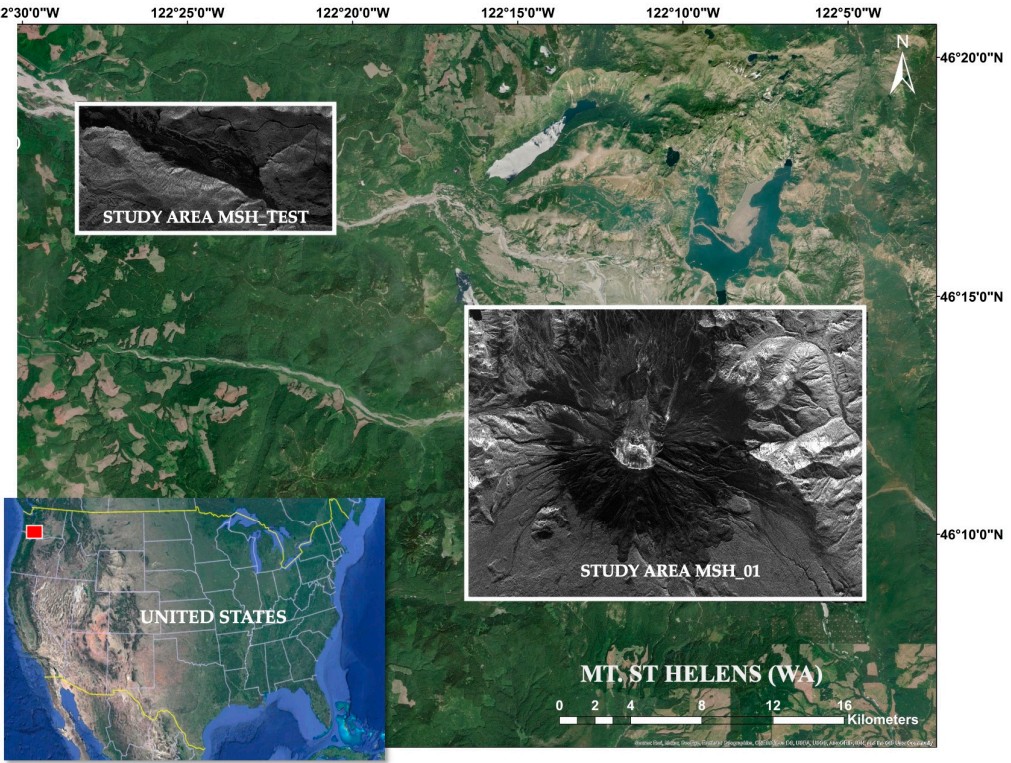

**Figure 1.** Context map of the two Mount St. Helens study areas termed 'MSH_TEST' and 'MSH_01' superimposed on the satellite basemap of the area. The (HV polarization) backscatter data from the ASAR-ISRO airborne SAR collection on 15–16 December 2019 are shown for both study areas.

Datasets were acquired during the first ASAR-ISRO flight campaign on 15–16 December 2019, covering the entire region surrounding Mount St. Helens. Specifically, we investigate two study areas, carefully chosen for the exceptional preservation of various

volcanic flow deposits from the May 1980 eruption, namely from debris avalanches (DA), lahars (LH), and pyroclastic density currents (PDCs).

Both areas selected for this study (i.e., 'MSH_TEST' and 'MSH_01') include a range of deposits from the 1980 eruption, as well as older lava flows (LF) on the south flank (Figure 2).

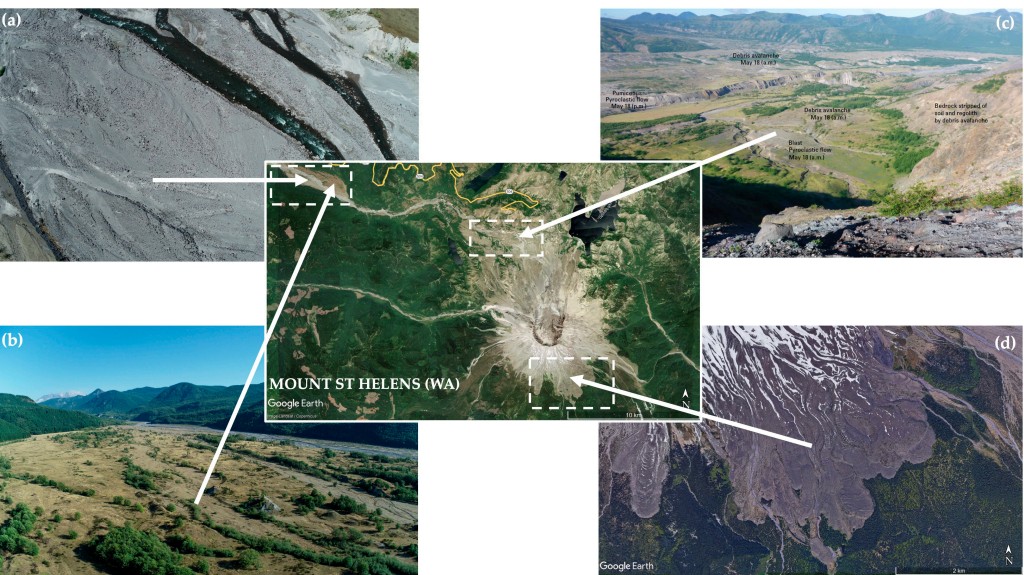

**Figure 2.** Mount St. Helens (WA) Google Earth base map (**center**) from 25 July 2021 (Landsat/Copernicus), showing localities of the deposits investigated [40], including images captured via UAS (during our field campaign in September 2021) of the (**a**) lahar deposits with active channel: centimeter- to meter-scale surface roughness, ash-rich matrix and no vegetation, (**b**) debris avalanche deposits cut by lahar channels: decameter-scale surface roughness, partially vegetated with hummocks > tens of meters. (**c**) Shows PDC deposits cut by lahar channels: centimeter- to meter-scale surface roughness, partially vegetated, ash-rich matrix with pumices > 1 m (from [40]), and (**d**) old lava flows (Google Earth): decameter-scale surface roughness, pahoehoe ropy texture (smooth), and a'a and blocky lavas at steep fronts, with sparse vegetation (on lower slopes) and snow cover (upper slopes).

The study area 'MSH_01' includes most of the deposit types representative of the 1980 eruption, as it covers the summit and the northern flank of the volcano. The May 1980 debris avalanche turned westward as far as 23 km down the valley of the North Fork Toutle River and formed the hummocky deposits investigated in the 'MSH_TEST' study area [39].

The deposit volume of the debris avalanche from the eruption is about 2.5 cubic kilometers, representing a dynamic landscape with a variety of deposits, for which the geological history has been previously characterized [39,41].

## *2.2. Roughness Discrimination*

Lahar deposits in the North Fork Toutle River are cut by a wet active channel in the 'MSH-TEST' study area. They show centimeter- to meter-scale surface roughness, with only sparse grasslands to no vegetation, and are poorly sorted with boulders > 1 m inside an ash-rich matrix (<2 mm size). Debris avalanche deposits in this area are cut by dry lahar channels. They exhibit decameter-scale surface roughness, are partially vegetated with trees and grasslands, and are very poorly sorted with hummocks > 10 s meters in size composed of hydrothermally altered blocks and ash-rich matrix.

In study area 'MSH_01', PDC deposits are also cut by dry lahar channels. They are characterized by centimeter- to meter-scale surface roughness, are partially vegetated by grasslands, and are poorly sorted with pumices > 1 m in size inside an ash-rich matrix (<2 mm size). Finally, lava flows are only present in study area 'MSH_01' and represent

decameter-scale surface roughness, with either pahoehoe ropy texture (smooth) or a'a texture (rough) with blocky steep fronts. They are sometimes covered by sparse vegetation (in lower slopes only) and snow (upper slopes only).

The Rayleigh criterion, defined as:

$$h > \frac{\lambda}{8\cos(\theta)} \tag{1}$$

where $\lambda$ is the band wavelength and $\theta$ is the incidence angle, provides a material size threshold of whether a surface appears 'rough' (bright) or 'smooth' (dark) in backscatter, using the same instrument characteristics and scattering properties of the ground. Using the range of local incidence angles mentioned above, for S-band, objects either <1.2 cm (incidence angle of 22°) or <5.0 cm (incidence angle of 77°) appear smooth, while for L-band objects < 3.6 cm or <13.3 cm will appear smooth. This means that we can expect material sizes between 1.2 cm and 13.3 cm to produce different backscatter signals at L- and S-band wavelengths.

*2.3. Roughness Computation*

NASA-ISRO provided an equation along with the Level 2 geocoded product data to compute backscatter. The radar backscattering signal is calculated for each pixel in the Level 2 geocoded product dataset by applying the equation:

$$\sigma^0 = 10\log\left(D^2 - N\right) + 10\log(\sin\alpha) - K_{dB}, \tag{2}$$

where $\sigma^0$ is the backscattering coefficient in dB, $D$ is the digital number, $N$ is the image noise bias, $\alpha$ is the incidence angle, and $K_{dB}$ is the calibration constant. The value of $N$ depends on the scene, polarization mode, and frequency as reported in supplemental information (Table S1). Importantly, pixels for which the image noise bias is greater than $D^2$ are deemed unreliable and assigned a 'NoData' value.

A scene-dependent pixel-specific weight, $I_i$, for pixel index $i$ is then applied to each coefficient to yield a new geocoded dataset, $\sigma\prime$, that contains "intensity-weighted" backscatter coefficients:

$$\sigma'_i = \sigma^0_i I_i, \tag{3}$$

where

$$I_i = \frac{D^*_i - \min(D^*)}{max(D^*) - \min(D^*)} \tag{4}$$

and

$$D^* = -D \tag{5}$$

The range of values $I_i$ can take on is constrained *a posteriori* by bounding the distribution at two standard deviations. Out-of-bounds values are adjusted so that they lie on the boundary. This dataset is then processed to yield another geocoded dataset, $\hat{\sigma}\prime$, in which any pixels containing 'NoData' are assigned a value, $\overline{W}^n$, equal to the average of their nearest neighbors using an $n \times n$ window centered on the pixel of interest. We ignore 'NoData' pixels when computing this average and define $\omega$ as the number of 'NoData' pixels contained within the window. In cases where all nearest neighbors contain 'NoData', such that $\omega = n^2$, the average is taken to be 'NoData'.

$$\hat{\sigma}'_i = \begin{cases} \sigma'_i, & \text{if} \sigma'_i \neq NoData; \\ \overline{W}^n, & \text{if } \omega < n^2; \\ NoData, & otherwise. \end{cases} \tag{6}$$

The per-pixel roughness, $\rho_i$, is finally computed as,

$$p_i = \frac{\sum_{j=0}^{n} \sum_{k=0}^{n} \left| W_{j,k}^n - \overline{W}^n \right|}{n^2}, \tag{7}$$

where $\rho_i$ is the mean absolute deviation (MAD) of backscattering coefficients, $\hat{\sigma}'$, for the set of pixels contained in an $n \times n$ window, $W^n$, centered on pixel $p_i$, and $j, k$ are the indices of the window's rows and columns, respectively. Alternatively, if the window, $W^n$, is found to contain a 'NoData' value, then $\rho_i$ is assigned a value of 'NoData'. As a last step, the roughness values are rescaled to a range of [0,1] using min-max normalization. We note that as a consequence of using a windowed approach, the resulting roughness map is effectively cropped by $\lfloor n/2 \rfloor$ pixels on each side (Figure S1 in supplementary materials). However, instead of reducing the dimensions of the output, we assign a 'NoData' value to these marginal pixels.

*2.4. Dual-Band Map Generation*

To facilitate the generation of dual-band roughness maps, we employed an in-house program (https://doi.org/10.5281/zenodo.7894225) that was elaborated from Section 2.3. This program, henceforth referred to as nfg, accepts a geocoded dataset as input, applies a specified transformation, e.g., Equation (7), using a variable window, and outputs one or more geocoded datasets.

In this work, we utilized a five-step protocol to generate dual-band roughness maps from L- and S-band Level 2 geocoded product datasets (Figure 3). First, we use nfg to transform L-S ASAR-ISRO Level 2 geocoded product data into weighted backscatter coefficients by applying Equation (3) to each pixel. Then, to fill in as many 'NoData' pixels as possible, we use nfg to apply Equation (6) to the weighted backscatter coefficients using a $3 \times 3$ window. Next, nfg is used with a $41 \times 41$ window to convert the backscatter coefficients into a measure of roughness according to Equation (7). Finally, nfg is used to split the roughness dataset based on a roughness cutoff of 0.5 resulting in two "split roughness" datasets, $\rho^+$ and $\rho^-$, where for a given pixel index, $i$,

$$\rho_i^+ = \begin{cases} \rho_i & if \, \rho_i \geq threshold; \\ NoData & otherwise, \end{cases} \tag{8}$$

and

$$\rho_i^- = \begin{cases} \rho_i & if \, \rho_i < threshold; \\ NoData & otherwise. \end{cases} \tag{9}$$

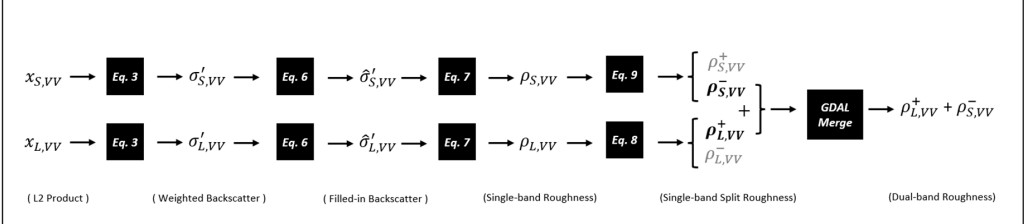

**Figure 3.** Dual-band map generation protocol. Starting with Level 2 geocoded product datasets, backscatter coefficients are computed using Equation (3) followed by an attempt to fill in missing data using Equation (6). Then, single-band surface roughness measurements are extracted from the backscatter coefficients using Equation (7). Next, the single-band roughness measurements are split into separate datasets using Equations (8) and (9). Dual-band roughness maps are finally constructed by combining two split datasets using GDAL Merge.

A dual-band roughness map is then constructed by combining two split roughness datasets with the GDAL Merge utility operating in mosaic mode. If overlapping pixels are

encountered during the merger, the L-band input is given priority, and the S-band input is discarded.

We determined window sizes based on the scale (size) of the volcanic mass flow features that we were trying to map and differentiate from one another. The $41 \times 41$ window, for example, was the best fit for mapping and differentiating the deposit characteristics of debris avalanches (hummocks with tens-of-meters-scale roughness) and lahars (blocks with meter-scale roughness) in the MSH_TEST area. The window size selection process was performed while maintaining concordance with corresponding volcanic mass flow deposits identified on the DEM [42] and geological map [43] of MSH while also considering surficial decadal changes (i.e., erosion and vegetation cover/removal).

## 3. Airborne SAR Data Calibration Using Unoccupied Aircraft Systems (UAS)

### 3.1. Calibration Methodology

We leveraged a DJI Phantom 4 RTK UAS, in combination with a differential GNSS base station (DJI D-RTK 2), to capture optical imagery across several swaths of the 'MSH_TEST' area during the 2 and 3 September 2021. We selected this area due to of the presence of debris avalanche deposits expressed through the substantial number of hummocks stemming from the 1980 eruption, offering an optimal geomorphological scenario for performing both qualitative and quantitative roughness calibration procedures.

We further investigated one UAS sub-swath, 'MSH_TEST_Survey_04', which exhibited the lowest presence of vegetation. We performed our flights at an altitude of 70 m in respect to the surface, with an overlap of 80 percent vertically and horizontally. We processed the UAS imagery using Metashape (Agisoft), a photogrammetry software based on the SfM method, to generate a dense point cloud from a total of 1198 aligned images. This dense point cloud was subsequently processed into a DSM with a 4.49 cm ground sample distance. This DSM resolution was far greater than the ASAR-ISRO resolution, resulting in our subsequent resampling of the UAS-derived DSM down to $2 \times 2$ m for both platforms to match, while still retaining some of the fidelity of the initial sub-meter UAS-derived products.

Through a preliminary qualitative examination of the ASAR-ISRO roughness outputs, we identified that the L-Band VV polarization provided the most adequate single-band roughness outputs to discern between deposits and therefore, be employed in our calibration procedure. Herein, we applied the first three steps of the dual-band map generation described in Section 2.3 to further analyze the chosen L-VV dataset using a series of windows ranging from $3 \times 3$ to $45 \times 45$ in $6 \times 6$ increments. This approach enabled us to evaluate the impact of window size on the roughness results, which may be further affected by flow and deposit type and, thus, ultimately specific to the target volcano.

By using a combination of the DSM and a tiled model generated via Metashape, along with the 'Imagery' basemap provided by ArcMap (Esri) containing optical imagery dated 28 June 2021 and Google Earth Pro optical imagery dated 7 September 2021 (Landsat/Copernicus), we visually identified 41 hummocks from which three were selected for further assessment: 'HM_05', 'HM_12', and 'HM_32' (Figure 4). These hummocks were selected based on apparent minimal vegetation coverage in the UAS-derived tiled model, while ensuring that their perimeters were fully contained within the bounds for all ASAR-ISRO roughness window sizes analyzed. Finally, to extract a ground-truth roughness measure from the $2 \times 2$ m DSM, we applied a $3 \times 3$ window and computed the mean absolute deviation of the elevation data.

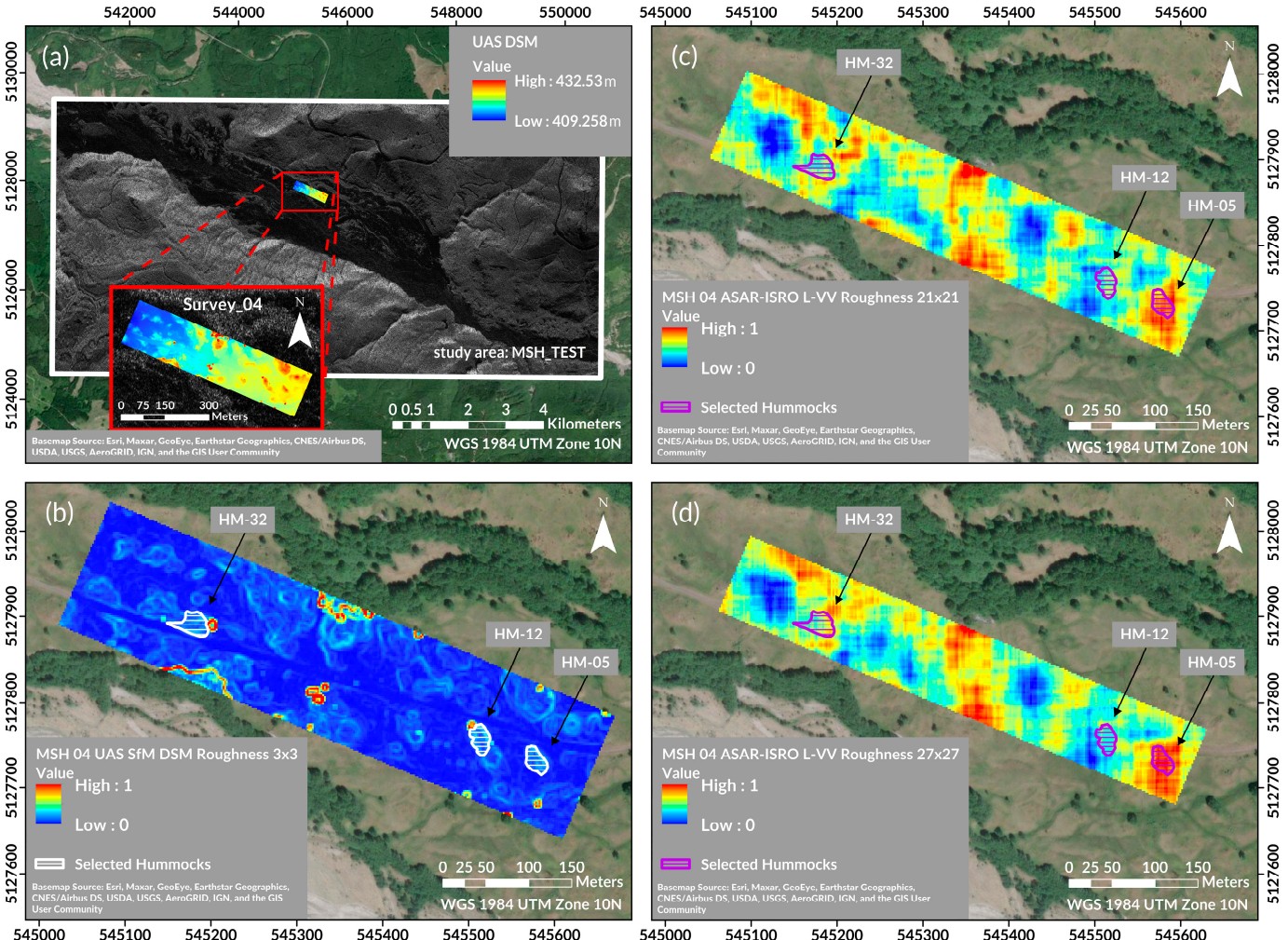

**Figure 4.** (**a**) Location of 'Survey_04' area within study area MSH_TEST—the background grayscale image is the ASAR-ISRO Level 2 geocoded product amplitude; (**b**) roughness map computed from resampled 2 m UAS SfM DSM using a 3 × 3 pixel window; (**c**,**d**) computed roughnesses for the ASAR-ISRO L-VV data, with 21 × 21 and 27 × 27 pixel windows, respectively. Pink outlines define the three hummocks analyzed.

Thereafter, we conducted a comparative analysis of the two roughness models obtained from the UAS DSM and ASAR-ISRO data. Specifically, we used the '3D Analyst' tool package within the ArcMap GIS (Esri) software to interpolate a long axis and a short axis line for each of the three hummocks selected. Subsequently, we extracted the corresponding roughness values along the interpolated lines using the 'Profile Graph' function and replicated this step for all eight windows (3 × 3 to 45 × 45) for the ASAR-ISRO data, while the 3 × 3 window was the only one used for the UAS DSM data. Finally, we examined the elevation data along the short and long axis of the UAS DSM to qualitatively cross-reference with the roughness results. This procedure resulted in a total of 60 profiles (6 elevation and 54 roughness) for this calibration.

To compare the two datasets, we employed a Gaussian kernel regression approach on the 54 roughness profile datasets. The purpose of this method was to achieve smooth curves while reducing the uncertainty of the spatial interval between each computed datapoint through proper interpolation. This technique allowed for a quantitative comparison between the UAS dataset and the ASAR-ISRO dataset, providing insights into the suitability of each model for hummock identification, roughness computation, and subsequent mapping.

The Gaussian kernel function $K(x_1, x_2)$ is defined as follows:

$$K(x_1, x_2) = exp\left(-\frac{|x_1 - x_2|^2}{2h^2}\right). \tag{10}$$

where $x_1$ and $x_2$ are two input vectors, $|x_1 - x_2|$ is the Euclidean distance between them, and $h$ is the bandwidth parameter.

Since the bandwidth parameter $h$ has a direct effect on the smoothed predicted curve, we used the k-fold cross-validation approach [44] to find the best bandwidth value ($h_{best}$) for the kernel regression. The function achieves $h_{best}$ by minimizing the mean of the mean squared errors over all the number of cross-validation splits,

$$h_{best} = argmin\, h \frac{1}{nsplits} \sum_{i=1}^{n_{splits}} \frac{1}{n_i} \sum_{j\in\, test_i} (y_j - \hat{y}_j)^2. \tag{11}$$

where $test_i$ is the test set for the $i-$th split, $n_i$ is the number of samples in the test set for the $i-$th split, $y_j$ is the true output value for the $j-$th sample in the test set, $\hat{y}_j$ is the predicted output value for the $j-$th sample in the test set using kernel regression with bandwidth $h$, and $nsplits$ is the total number of splits used in cross-validation.

Furthermore, to visualize the uncertainty associated with the predicted lines $\hat{y}_j$, we applied asymptotic variability bounds in the form of 95% confidence intervals. We then quantified the differences between the predicted lines obtained from the ASAR-ISRO roughness at each window size and the UAS DSM roughness at a $3 \times 3$ window by computing (i) the ratio of their standard deviations, where the ratio referred to the process of calculating the standard deviation for each, the ASAR-ISRO Gaussian kernel regression and the UAS Gaussian kernel regression, to then proportionally assess them in relation to one another; and (ii) their mean absolute errors, calculated specifically on the trendline of the ASAR-ISRO Gaussian kernel regression in respect to the trendline of the UAS Gaussian kernel regression. Our long and short axes profiles for the Gaussian kernel regression results of the selected hummocks and their related statistical outputs can be found in Figures S2–S4 (supplementary materials).

### 3.2. Calibration Results

Our results showed that when compared against the UAS DSM $3 \times 3$ roughness, either the $21 \times 21$ or the $27 \times 27$ window for the ASAR-ISRO roughness model produce the lowest ratios of standard deviation. This suggests that these windows allow for a similar range of roughness values to those obtained from the UAS DSM (Figure 5). Similarly, a lower ratio of standard deviations between the UAS and ASAR-ISRO roughness predicted lines aligned to a lower mean absolute error in many instances, which suggests that (i) the two datasets have similar trends and (ii) the ASAR-ISRO roughness outputs are likely to be more reliable. Conversely, there were exceptions, where sometimes smoothing the curve slightly increased the mean absolute error, as seen in 'HM_0'5 (Figure 5). This exception as an inverse relationship was most noticeable in cases where the trends of the predicted lines closely matched each other, yet their values did not converge at any point (Figure 5).

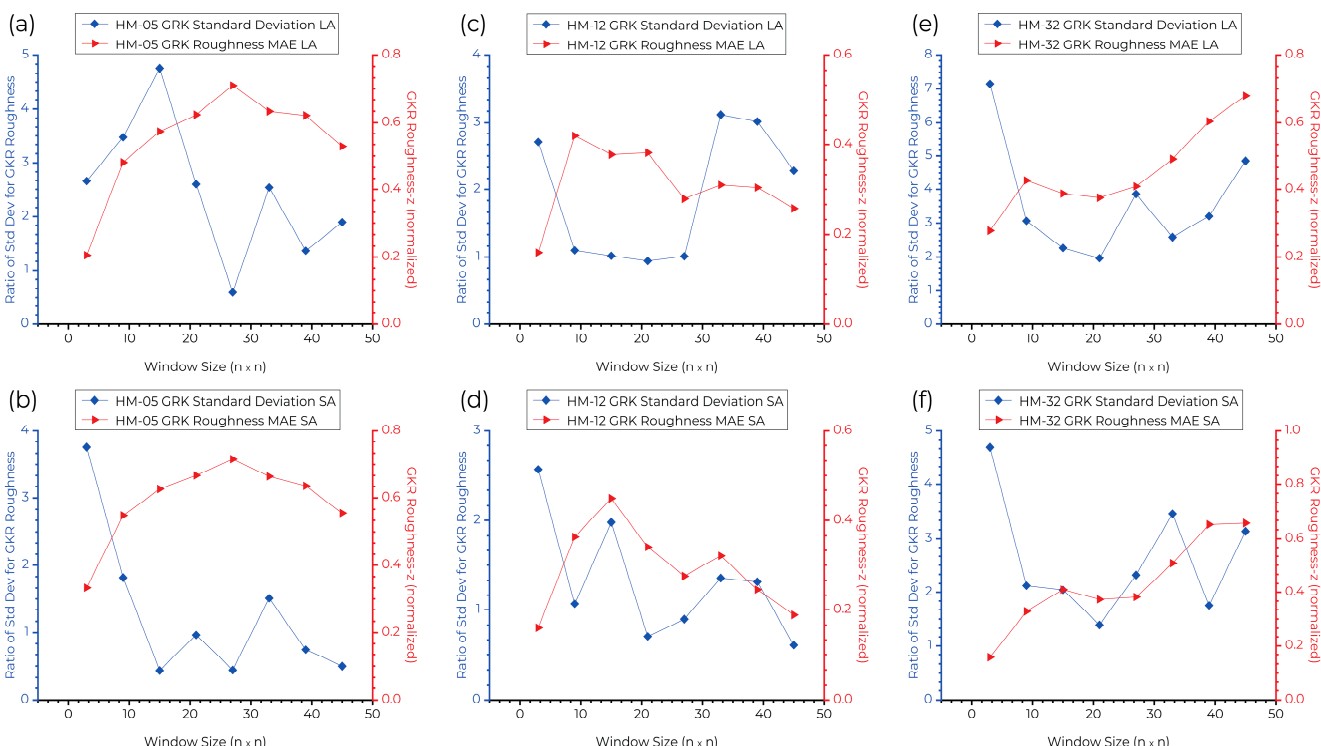

**Figure 5.** Plots for the ratio of standard deviation (blue) and mean absolute error (red) between predicted lines computed using Gaussian kernel regression on the UAS DSM and ASAR-ISRO datasets. The ASAR-ISRO datasets were processed through eight window sizes and compared against the UAS DSM 3 × 3 window dataset, where (**a**,**b**) corresponds to 'HM_05', (**c**,**d**) to 'HM-12', and (**e**,**f**) to 'HM_32'.

## 4. Results

### 4.1. ASAR-ISRO Backscatter Results

Following the methodology detailed in Section 2.3, we produced 2 m spatial resolution, georeferenced, and terrain-corrected ASAR-ISRO backscatter maps for all four datasets (Figure 6) over the target area (i.e., Mount St. Helens volcano). The data processing was carried out on 32 L-S ASAR-ISRO Level 2 geocoded datasets. The minimum and maximum backscatter values (i.e., range) for each dataset is given in two wavelengths (i.e., 24 cm and 9 cm) and four available polarizations, i.e., HH, HV, VH, and VV, indicating the backscattering characteristics of such dual-band datasets.

To identify and characterize the extent of the 18 May 1980 debris avalanche (DA) and syn- to post-eruptive lahar (LH) deposits, we used the 'MSH_TEST' study area, showing single-polarization backscatter maps (Figure 7 and Figure S5 in supplementary materials), as well as polarimetric decomposition backscatter maps (Figure 8).

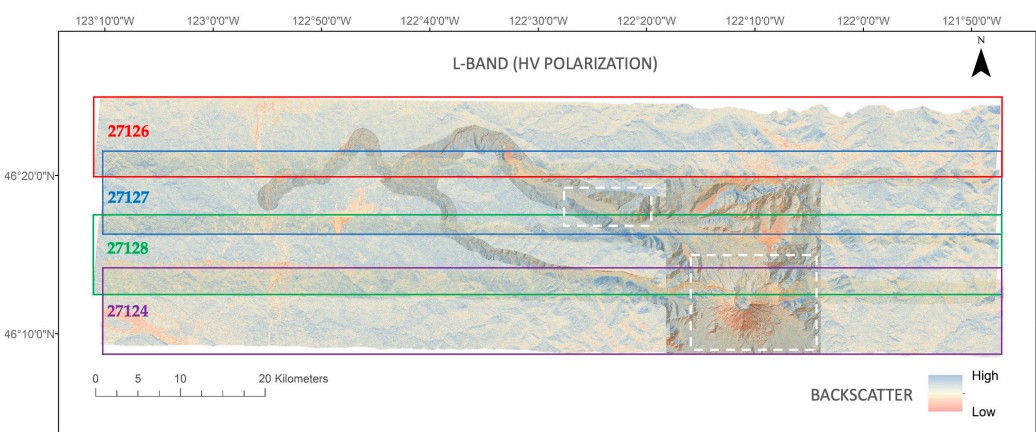

**Figure 6.** An example showing L-band backscatter signal (sigma-0, Equation (2)) in HH polarization for the ASAR-ISRO datasets (strips) over the region of Mount St. Helens volcano, superimposed on a 3 m LiDAR digital elevation model (DEM) of Mount St. Helens [42]. Minimum and maximum backscatter values are reported in Table 1.

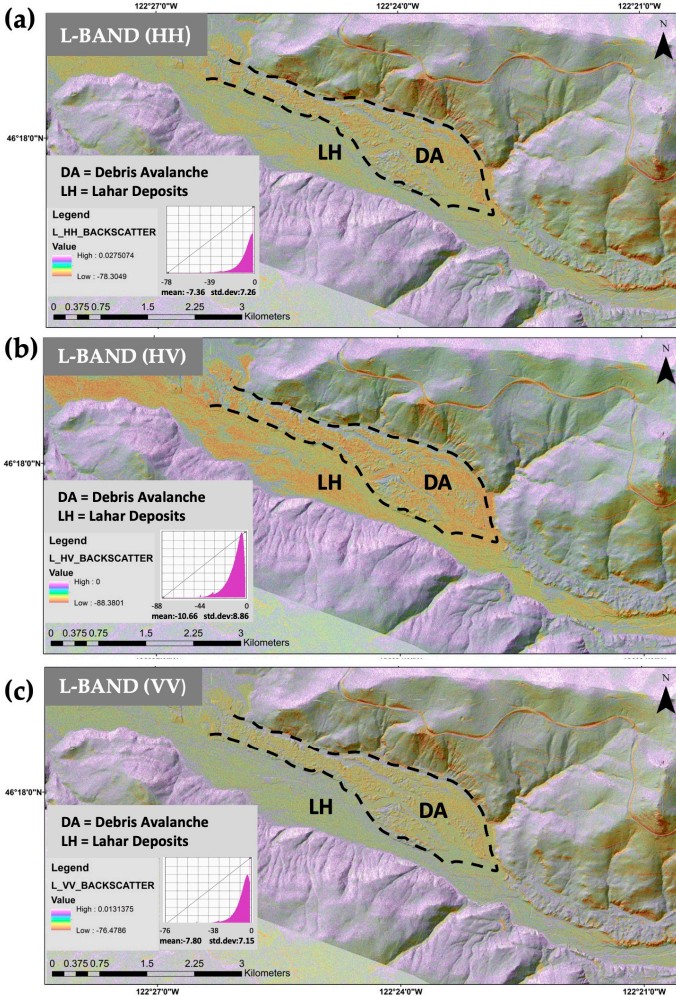

**Figure 7.** 'MSH_TEST' study area backscattering maps in L-band, showing a measurable backscatter variation in (**a**) HH, (**b**) HV, and (**c**) VV polarization. For visual clarity, ASAR-ISRO backscattering data were superimposed on a 3 m LiDAR DEM of Mount St. Helens [42].

**Table 1.** Backscatter signal (dB) obtained for all ASAR-ISRO datasets.

| STRIP ID | DATE/TIME (PST UTC -8) | | L-BAND POLARIZATION MODE | | | | S-BAND POLARIZATION MODE | | | |
|---|---|---|---|---|---|---|---|---|---|---|
| | | | HH | HV | VH | VV | HH | HV | VH | VV |
| 27124 | 16 December 2019 09:56 | Min | 41.5 | 26.6 | 26.3 | 38.5 | 41.0 | 22.9 | 24.3 | 36.1 |
| | | Max | −97.4 | −112.4 | −107.8 | −102.5 | −100.5 | −101.8 | −101.2 | −97.3 |
| 27126 | 15 December 2019 13:25 | Min | 30.5 | 20.3 | 20.0 | 30.1 | 32.6 | 17.2 | 18.1 | 33.1 |
| | | Max | −98.4 | −99.4 | −98.6 | −94.8 | −94.3 | −100.7 | −97.1 | −90.2 |
| 27127 | 15 December 2019 13:54 | Min | 33.6 | 19.0 | 18.5 | 32.9 | 37.1 | 19.6 | 20.2 | 38.6 |
| | | Max | −96.4 | −106.0 | −111.0 | −109.5 | −104.7 | −103.9 | −103.8 | −91.6 |
| 27128 | 15 December 2019 14:28 | Min | 34.0 | −18.0 | 16.9 | 30.7 | 41.6 | 27.6 | 28.1 | 35.0 |
| | | Max | −117.3 | −101.8 | −114.6 | −95.5 | −101.1 | −100.8 | −105.7 | −97.6 |

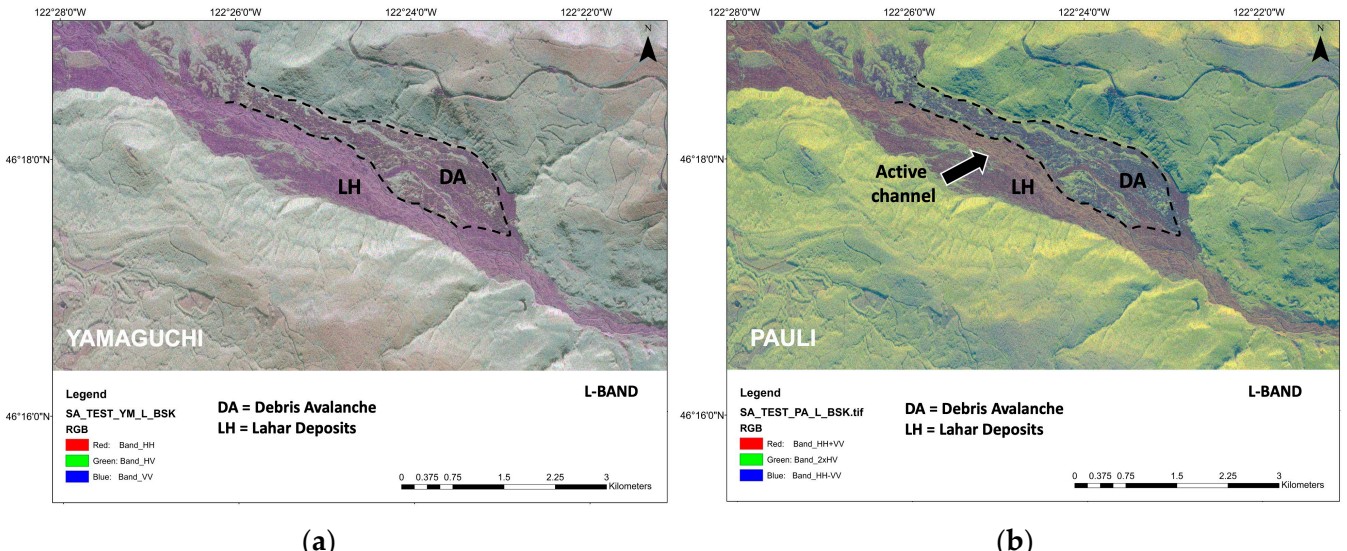

(**a**)                    (**b**)

**Figure 8.** 'MSH_TEST' study area backscatter data, superimposed on a 3 m LiDAR DEM of Mount St. Helens [42], where (**a**) shows the L-band Yamaguchi polarimetric decomposition map (RGB: HH, HV and VV respectively) and (**b**) shows L-band Pauli polarimetric decomposition map (RGB: HH+VV; 2 × HV and HH-VV, respectively) of the same target area. The 1980 eruption debris avalanche deposits (DA) are indicated with a black dashed line, bordering some lahar deposits (LH) to the south of the valley, where the active channel (indicated by a black arrow) is also detectable.

Figure 7 shows that using the L-band dataset in HV polarization (Figure 7b) has the strongest signal to highlight DA deposits. However, co-polarized data in L-band can better distinguish the different radar backscatter signals obtained between the DA and LH deposits. Furthermore, backscatter data obtained with the S-band (see Figure S5d–f in supplementary materials) does not appear to be as suitable for distinguishing these two different deposits; hence, we prioritize L-band data for the roughness computation.

Polarimetric decomposition methods [45] allow the separation of different scattering contributions and are used to extract information about the scattering process that characterizes the backscatter signal obtained over each type of deposit. For example, it may be used to mask the heavily vegetated areas, which would highlight the area of interest (DA and LH deposits), as well as make the active channel more prominent (Figure 8).

This is also in agreement with a previous claim that polarimetric decomposition techniques have been successfully used to separate and remove the disturbing vegetation contribution and allow estimation of the soil moisture content of the isolated surface components [46].

In contrast, the 'MSH_01' study area backscatter maps (Figure S6 in supplementary materials) show more complex amalgamation of volcanic deposits (Figure 9) from lava flows on the southern flank of the volcano summit (outlined in red, LF), an extensive network of lahar (outlined in yellow, LH), and PDC deposits on the northern flank (outlined in white, PDC). Employing the same approach as for the 'MSH_TEST' study area (Figures 7 and 8), it is evident that the single-polarization maps for the 'MSH_01' study area show variable backscatter intensity in both cross-polarized and co-polarized acquisitions (Figure S6 in supplementary materials).

Nonetheless, the two polarimetric decomposition methods used here allow the separation of different scattering contributions (Figure 9) by masking vegetated areas and highlighting exposed terrain associated with LF, LH, or PDC deposits. Single-bounce scattering is dominant over the pristine 1980 volcanic deposits, which contrasts with the surface and volume scattering that characterize older, vegetated (and/or snow-covered) terrains. However, the backscatter values for all three deposits analyzed are of similar intensities and cannot be distinguished visually with absolute certainty. For this reason, we attempted a statistical approach to differentiate the four different flow deposits studied, based first on their backscattering characteristics.

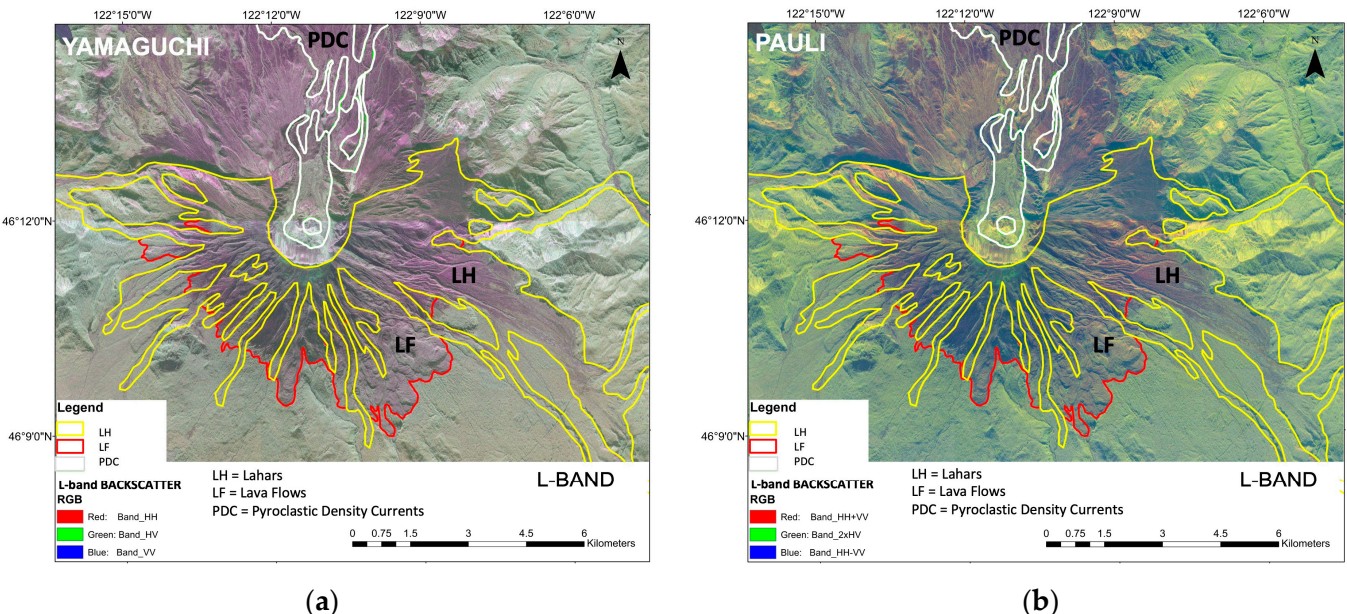

(**a**)           (**b**)

**Figure 9.** 'MSH_01' study area backscatter data superimposed on a 3 m LiDAR DEM of Mount St. Helens [42], where (**a**) shows Yamaguchi polarimetric decomposition map and (**b**) shows the Pauli polarimetric decomposition map. The extent of LH, LF, and PDC deposits (extracted from [39,43]) are superimposed on backscattering results and are indicated in yellow, red, and white, respectively.

Focused Studies: Backscatter Results

To correctly analyze and interpret the backscatter signal extracted from the ASAR-ISRO data over the 'MSH_TEST' study area, prior knowledge of the different volcanic deposit types and land cover present were needed. We extracted polygons as digital representations for each volcanic deposit type analyzed, from the post-1980 geological map [39], which we collected during our field investigation at Mount St. Helens (WA) in September 2021 (Section 3). This was necessary to account for the 40+ years of erosion and vegetation cover that have affected the surface of the 1980 deposits. We note that since the ASAR-ISRO data we obtained was collected in December 2019, we could not directly assess in the field the extent of the active channel and/or possible snow cover in the 'MSH_TEST' area at the time of data acquisition. See Sections 4 and 5 for details.

By implementing a focused study for the 'MSH_TEST' study area, we attempted a statistical differentiation of DA from LH deposits by processing the L-band datasets (which

are superior for such differentiation; see previous section) with a 'thresholding' method, where backscatter values between the two distinctly different deposits (i.e., DA and LH) would not overlap. An example of this approach is shown in L-band HV polarization (Figure 10), as deposit differentiation has been successfully achieved in that polarization alone, where backscatter values vary between −48.05 and −19.69 for LH deposits and 2.17 and −20.33 for DA deposits (backscatter values in bold for L-band HV polarization).

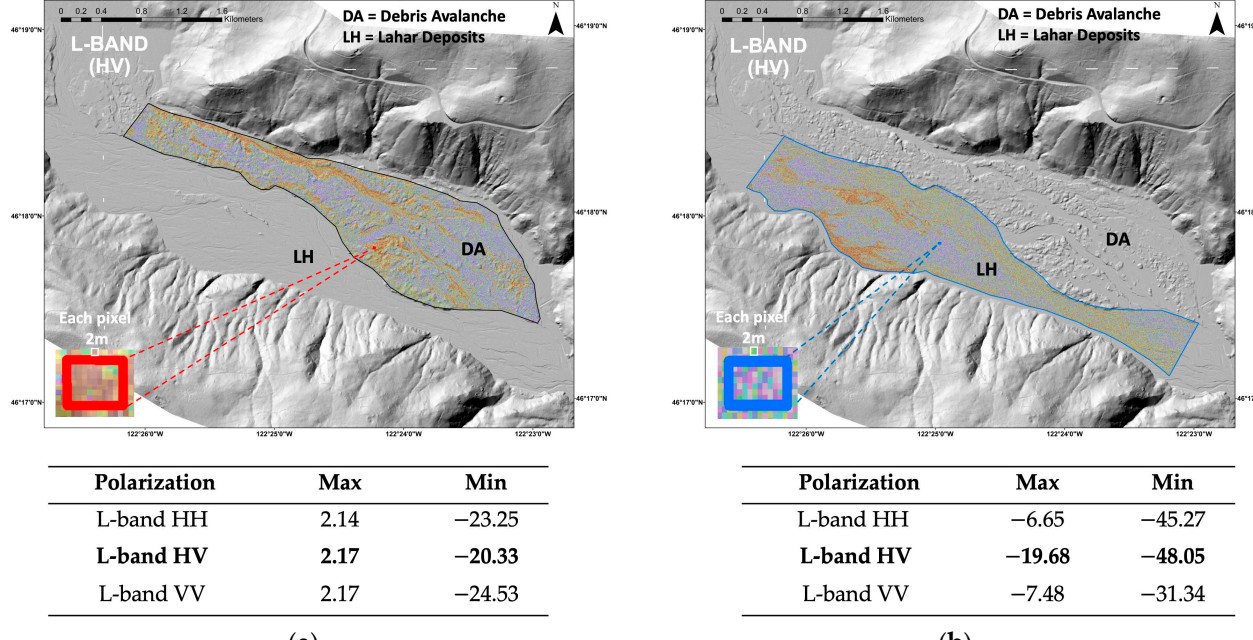

| Polarization | Max | Min |
|---|---|---|
| L-band HH | 2.14 | −23.25 |
| **L-band HV** | **2.17** | **−20.33** |
| L-band VV | 2.17 | −24.53 |

(**a**)

| Polarization | Max | Min |
|---|---|---|
| L-band HH | −6.65 | −45.27 |
| **L-band HV** | **−19.68** | **−48.05** |
| L-band VV | −7.48 | −31.34 |

(**b**)

**Figure 10.** 'MSH_TEST' study area backscatter maps: (**a**) for debris avalanche (DA) deposits (red tones, high backscatter) and (**b**) for lahar (LH) deposits (blue tones, low backscatter). Backscatter range values for each deposit (i.e., DA and LH) at small scale (inset 10 × 10 pixels) are reported below each figure for co-polarized (HH and VV) and cross-polarized (HV) modes. Backscatter data in L-band and HV polarization shown in this example are superimposed on a 3 m LiDAR DEM of Mount St. Helens [42].

Similarly, for study area 'MSH_01', the extent of the three types of volcanic flow deposits investigated (Figure 11) reporting backscatter range (Tables 2 and 3) and type of scattering identified for each band and polarization are summarized in Table 3.

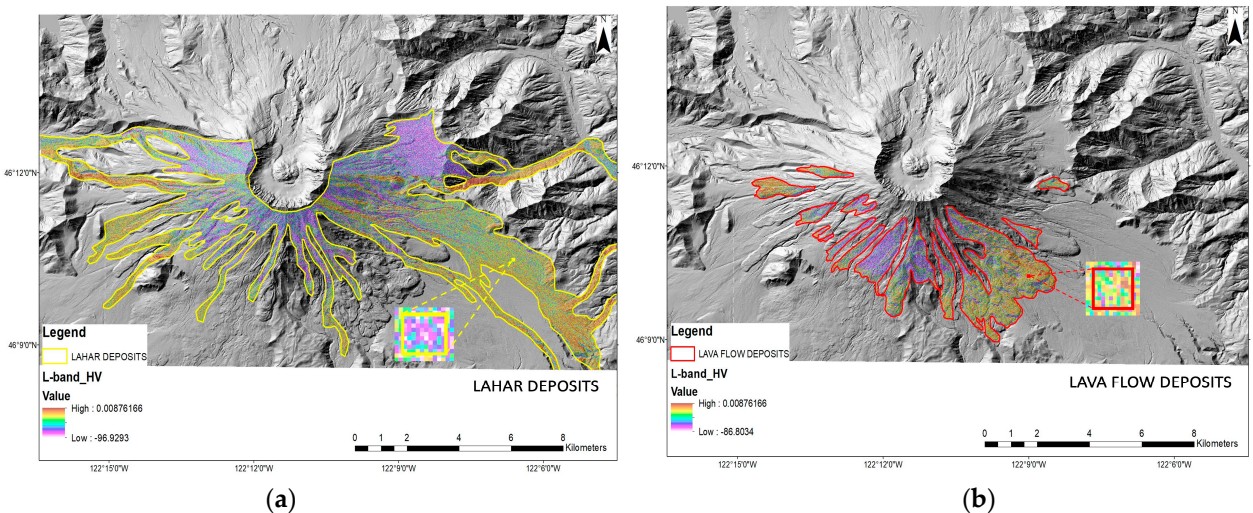

(**a**)

(**b**)

**Figure 11.** *Cont.*

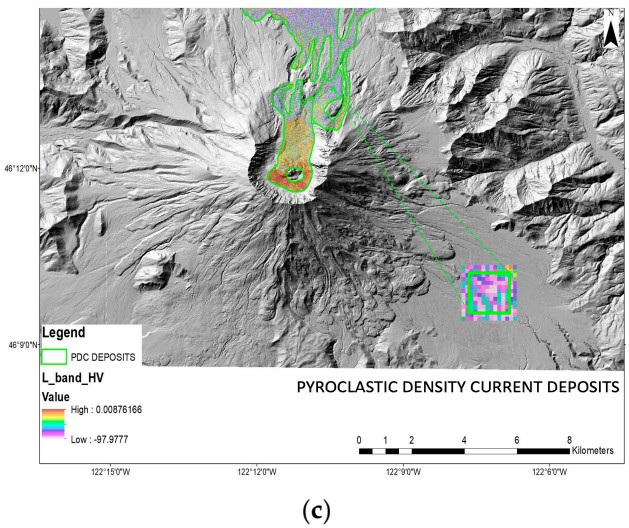

(**c**)

**Figure 11.** Focused study of volcanic flow deposits investigated in study area 'MSH_01' in L-band and HV polarization, which only show typical backscatter range for (**a**) lahar deposits, (**b**) lava flows, and (**c**) PDC deposits. Backscatter data are superimposed on a 3 m LiDAR DEM of Mount St. Helens [42].

**Table 2.** 'MSH_01' Deposits Backscatter Range *.

| Deposits | Max | Min |
|---|---|---|
| (a) Lahars | −20.9 | −57.1 |
| (b) Lava Flows | −15.6 | −53.0 |
| (c) PDCs | −25.2 | −62.8 |

* Expected (typical) backscatter range values (minimum and maximum) for each type of deposit at small scale (inset 10 × 10 pixels) are reported in Table 2. The range for the entire individual deposit is shown in Figure 11 and Table 3.

**Table 3.** Backscatter range (max/min) for type of individual volcanic deposit analyzed (extracted from 'MSH_01' backscatter data).

| Type of Deposit | Sensor Band * | Polarization | Backscatter Range (Actual) | Type of Scattering ** |
|---|---|---|---|---|
| Lahars | L-band | HH | Max: 0.0 Min: −90.6 | SB |
| | | HV | Max: 0.0 Min: −96.9 | SB |
| | | VV | Max: 0.0 Min: −85.3 | SB |
| | S-band | HH | Max 0.1 Min −83.1 | SB |
| | | HV | Max: 0.0 Min: −95.8 | SB |
| | | VV | Max: 0.1 Min: −83.5 | SB |
| Lava Flows | L-band | HH | Max: 0.0 Min: −96.8 | SB |
| | | HV | Max: 0.0 Min: −86.8 | SB |
| | | VV | Max: 0.0 Min: −82.6 | SB |
| | S-band | HH | Max: 0.1 Min: −100.5 | SB |
| | | HV | Max: 0.00 Min: −98.0 | SB |
| | | VV | Max:0.0 Min: −79.1 | SB |
| Pyroclastic Density Currents | L-band | HH | Max: 0.0 Min: −75.0 | SB |
| | | HV | Max: 0.0 Min: −98.0 | SB |
| | | VV | Max: 0.0 Min: −91.7 | SB |
| | S-band | HH | Max: 0.1 Min: −77.2 | SB |
| | | HV | Max: 0.0 Min: −84.5 | SB |
| | | VV | Max: 0.1 Min: −81.0 | SB |

* L-band and S-band 24 cm wavelength and 9 cm wavelength, respectively ** Single-bounce (SB) scattering. Minimum and maximum backscatter values for the individual deposit areas are reported in Table 2, detailing polarizations and types of scattering.

Expected or typical range of backscatter values for LH, LF, and PDC deposits reported in Table 2 using the L-band data (HV polarization) are small-spatial-scale (i.e., focused) range values (i.e., 10 × 10 pixels), which differ measurably from the range of backscatter values extracted from the entire, large-scale individual deposit areas (Figure 11 and Table 3). For example, the focused backscatter values obtained for lava flows (Figure 12b inset) is between −15.6 and −53.0 (Table 2), whereas the backscatter values obtained for the entire lava flow field (Figure 11b main) is between 0.0 and −86.8 (Table 3).

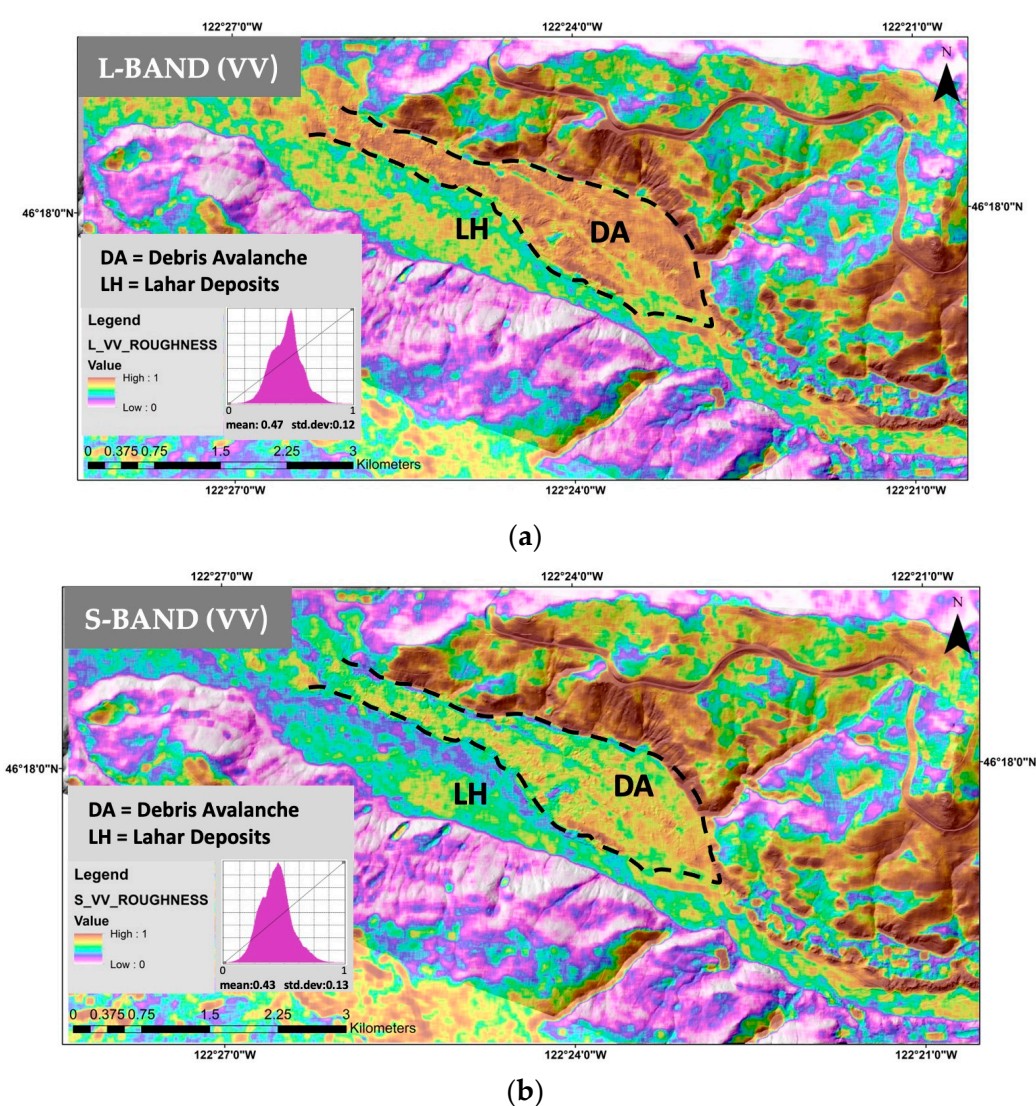

(**a**)

(**b**)

**Figure 12.** Surface roughness maps for 'MSH_TEST', derived from ASAR-ISRO backscatter data, using Equation (7) with a 41 × 41 window, superimposed on a 3 m LiDAR DEM of Mount St. Helens [42], where (**a**) is the L-band and (**b**) is the S-band, shown in co-polarized (VV) mode.

While some variations in the extracted backscatter signal are apparent when using the entire footprint of each deposit type with different band frequencies and polarization, the statistical approach proposed here was not successful for the 'MSH_01' study area (at large or small scale), as backscatter values between three deposit types overlap (Table 3), and an alternative approach using surface roughness instead of backscatter must be used.

### 4.2. ASAR-ISRO Surface Roughness Results

Using the methodology outlined in Section 2.3, we produced cross-polarized (HV) and co-polarized (HH and VV) surface roughness maps with the L-band (Figures 12a and S7a–c in supplementary material) and S-band (Figures 12b and S7d–f in supplementary material)

datasets over the 'MSH_TEST' study area. To visualize these maps, as well as roughness maps covering MSH_01, and to maximize the visibility of variation therein, we employed ArcMap and used histogram-equalized color ramps bounded 0 and 1. Surface roughness at wavelengths larger than the original L- and S-band wavelengths may be desirable when considering the application of larger windows for distinguishing various volcanic mass flow deposits.

Results show that the roughest visible terrain is on the northern riverbank of the North Fork Toutle River (WA), covered by the May 1980 DA deposits (brown tones). Dry lahar channels are visible, cutting through the DA deposits in this area (green linear features), in turn making the originally rough terrain smoother. We would expect fresh, hummocky DA deposits to be characterized by high roughness values (brown tones) within the 18 May 1980 DA outline (dashed black line). This correlates with the existing geological mapping [39], which describes syn- and post-eruptive lahar deposits covering the 18 May 1980 DA deposits in this area shortly after their emplacement [39].

Interestingly, the surface roughness results for the 'MSH_TEST' study area computed using Equation (7) (Figure 12) suggest that the best deposit differentiation between DA and LH is achieved using co-polarized (HH and VV) ASAR-ISRO data, which is in line with our backscatter findings and interpretation (Figure 7). Using co-polarized L-band data (Figure 12a and Figure S7a,c in supplementary materials), DA deposit roughness (brown tone indicating high surface roughness) is apparent and easily distinguished from LH deposit roughness (green tone indicating lower surface roughness). The old, vegetated terrain (purple tones indicating very low surface roughness) of the southern valley bank, bordering the LH area in the southernmost terraces of the riverbank, is evident using both L- and S-band datasets.

The S-band data (Figure 12b and Figure S7f in supplementary materials) seems to be less sensitive to the change of deposit surface roughness (for lahars and debris avalanches only) than the L-band data. Since hummocks in DA deposits are of large scale (>a few meters), they are better resolved using a longer wavelength (i.e., L-band) as opposed to the shorter S-band wavelength. Nonetheless, the S-band data allowed us to discern different terrain soil moisture conditions and identify the current wet active channels (blue tone indicating low surface roughness) that cut through the LH deposits. Considering that our surface roughness data for the 'MSH_TEST' area (Figure 12 and Figure S7 in supplementary materials) better differentiate the DA and LH deposits using co-polarized (e.g., VV) data, we applied the same approach to the 'MSH_01' study area using that polarization alone (Figure 13).

The deposits investigated in the 'MSH_01' study area (i.e., LF, LH, and PDCs) cover a significantly larger area than the deposits assessed in the 'MSH_TEST' area and have potentially more comparable (less distinctive) surface roughness than the DA and LH deposits. Therefore, the surface roughness results using L- and S-band data (Figure 13) may not correlate as well with the deposit footprints as those seen in Figure 12. Nonetheless, there are some significant and measurable roughness characteristics visible in S-band (Figure 13b) for the PDC deposits on the northern flank (purple tones indicating very low surface roughness), which may not be as apparent in L-band (Figure 13a). While LH deposits show variable surface roughness on the steep slopes on the southern flank where they overlay the lava flows, they are mostly characterized by intermediate-to-low surface roughness on lower slopes, especially on the southeastern flank (green tones indicating intermediate surface roughness), where they contrast with the old vegetated eastern volcanic features of Mount St. Helens (purple tones indicating very low surface roughness). The brown tones indicating higher surface roughness can be identified at the frontal lobes and edges of lava flows (LF) in both bands. Finally, some elongated topographic features mostly oriented east–west, also known as parallel to the line of sight (LOS) of the ASAR-ISRO radar during acquisitions, are clearly affected by radar shadowing and layover, where surface roughness appears artificially low on the northern sides of such

features directly exposed to the LOS and rough on the opposite southern sides (i.e., regions located west and east of the Mount St. Helens' edifice on Figure 13).

To refine our surface roughness method (Figures 12 and 13) and highlight specific surface features of such volcanic flow deposits that could have been hidden when using single-band roughness maps, we simultaneously processed the S-band and L-band data to produce a combined, dual-band surface roughness map, as described in Section 2.4.

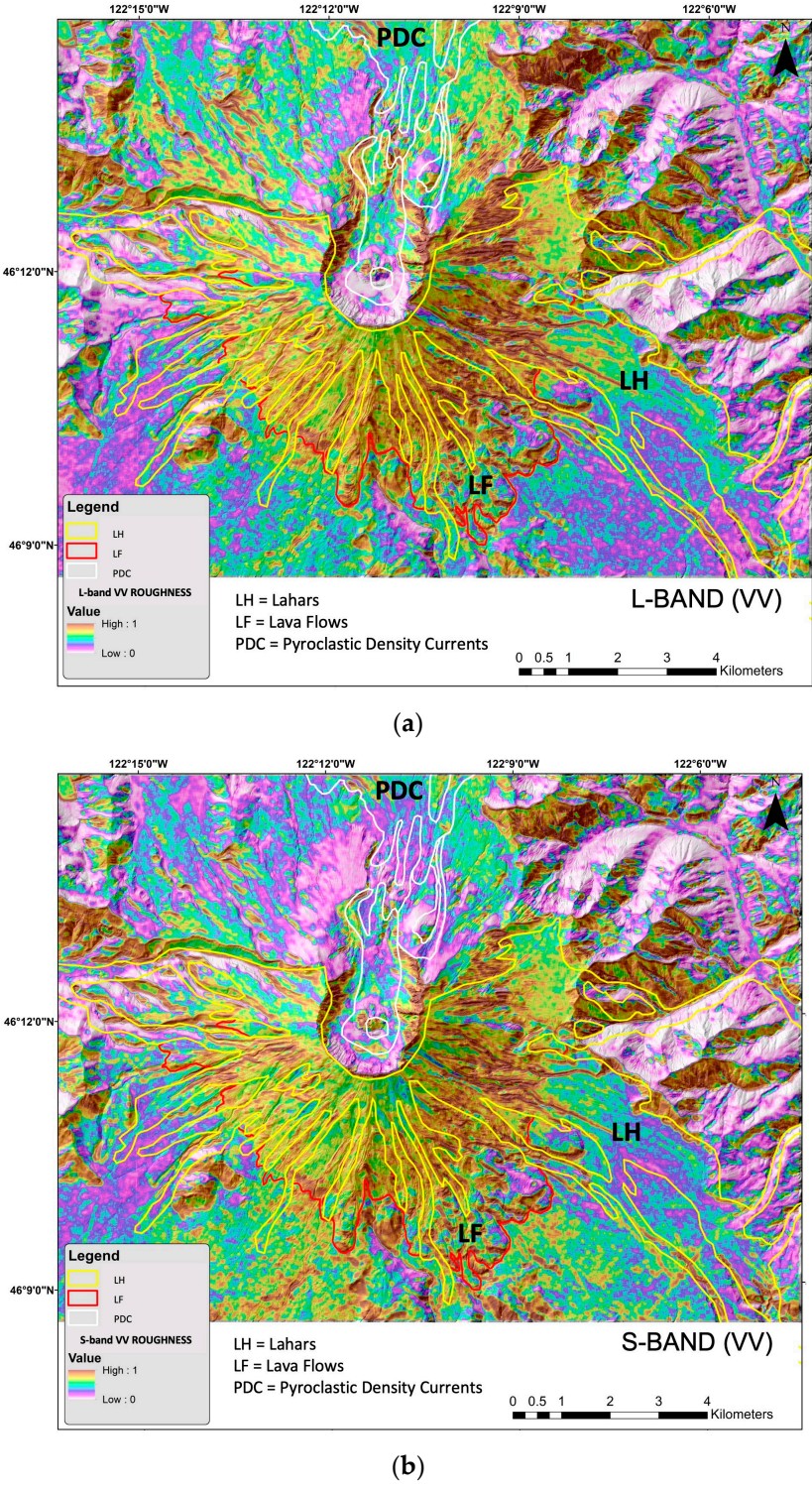

**Figure 13.** Surface roughness map for 'MSH_01', derived from ASAR-ISRO backscatter, using Equation (7) with a 41 × 41 window, superimposed on a 3 m LiDAR DEM of Mount St. Helens [42], (**a**) L-band and (**b**) S-band, both in VV polarization mode.

Dual-Band Surface Roughness Results

Our dual-band approach successfully combined two-band data (i.e., L- and S-band, both in VV polarization), producing a single L + S dual-band surface roughness map, which significantly improves the identification and differentiation of the four types of volcanic flow deposits studied for both the 'MSH_TEST' (Figure 14) and 'MSH_01' study areas (Figure 15) by accentuating the visual intensity of both high roughness values inside the L-band data and low roughness values from the S-band data.

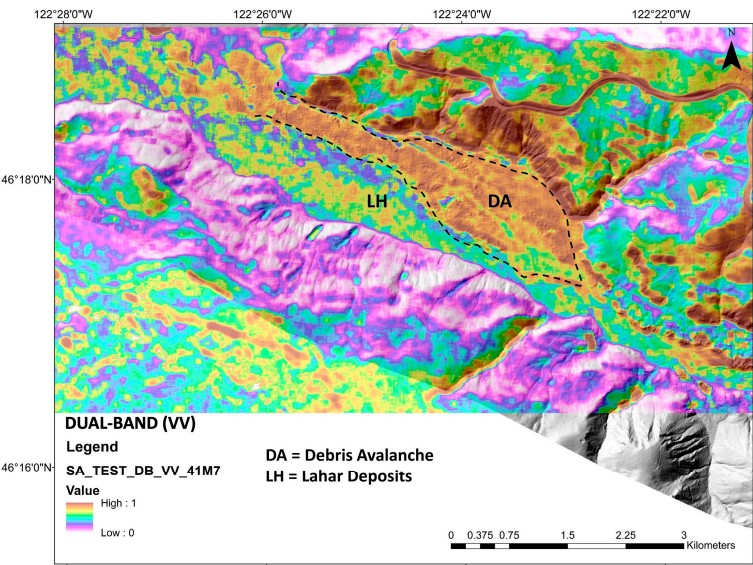

**Figure 14.** Surface roughness map combining L- and S-band data using a dual-band approach (detailed in Section 2.4) of the 'MSH_TEST' study area in VV polarization mode. This approach accentuates the visual intensity of both high surface roughness values inside the L-band data (e.g., debris avalanche deposits) and low (smooth) surface roughness values from S-band data (e.g., lahar deposits), including the active channel within the lahar deposits, otherwise undetectable in the L-band data. Surface roughness data are superimposed on a 3 m LiDAR DEM of Mount St. Helens [42].

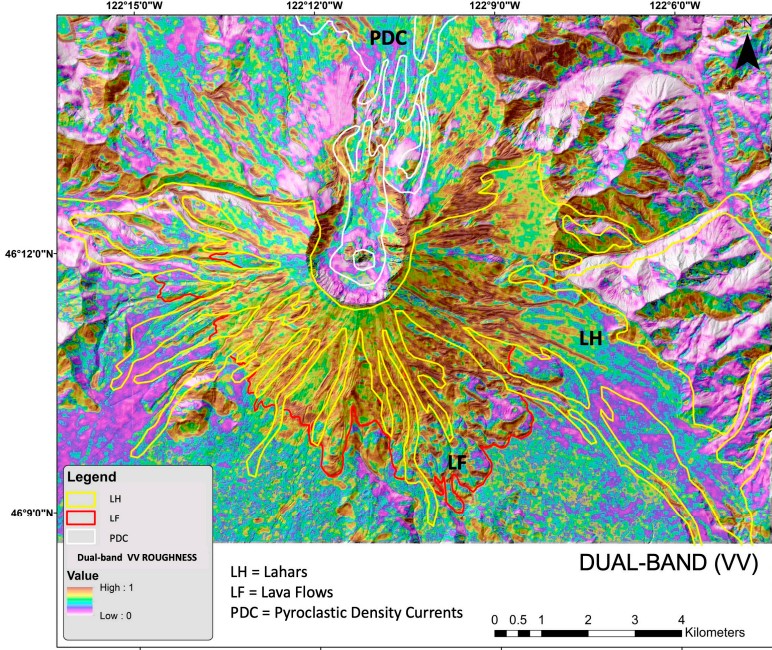

**Figure 15.** Surface roughness map combining L- and S-band data using a dual-band approach for study area 'MSH_01' in VV polarization mode. Surface roughness data are superimposed on a 3 m LiDAR DEM of Mount St. Helens [42].

The DA deposits in Figure 12 (brown tones indicating high surface roughness), most prominently identifiable in the L-band data (Figure 12a), are now complemented by a distinct footprint of LH deposits (green tones indicating lower surface roughness), also including the discernable wet channel area (blue tone indicating very low surface roughness) streaming through the lahar terraces, which was previously only distinguishable in S-band (Figure 12b). Moreover, the intermediate surface roughness of dry lahar channels cutting through the rough DA deposits are also visible (green to blue tones inside the DA footprint on Figure 14).

Similarly, the more complex amalgamation of volcanic deposits that covered the 'MSH_01' study area (Figure 15) are somewhat better resolved using the dual-band roughness approach, as evidenced by the better contrast displayed between the apparent rough surface (brown tones) of lava flows (LF) on the southern flank, and the low to intermediate roughness (green to purple tones) of recent PDC and lahar (LH) deposits on the northern and eastern flanks, respectively.

## 5. Discussion

### 5.1. Surface Roughness

We demonstrated in this study that mapping volcanic flow deposits using ASAR-ISRO remote sensing data can be achieved by utilizing different backscattering characteristics as a metric of surface roughness. To develop a robust method that does not rely on visual identification alone, we performed statistical tests of the backscatter results obtained for various deposit types. Results show that backscatter values for poorly sorted, fine-grained (with blocks < a few meters) deposits, such as lahars and PDCs, overlap; hence, they cannot be used for unique identification. Results obtained over the 'MSH_TEST' study area, using a small spatial scale ('focused') approach by solely extracting the most characteristic pixel values for each deposit, show that it is feasible to distinguish between debris avalanche deposits characterized by high backscatter values (from 2.17 to −20.33) and lahar deposits with low backscatter values (from −19.68 to −48.05). However, the statistical approach proposed here was not successful for the 'MSH_01' study area, and an alternative approach using surface roughness instead of backscatter only must be applied to such datasets.

When coupled with surface roughness, as derived here from the mean absolute deviation of radar backscatter, some target features can be confidently mapped and characterized (e.g., the 'MSH_TEST' study area).

Some flow deposits may display similar backscattering characteristics, such as PDCs and lahars, due to similar sorting and grain size distributions and are therefore difficult to distinguish statistically when present in the same study area (e.g., 'MSH_01'). Therefore, we suggest implementing focused study analyses like the ones shown here to determine the expected typical values for each deposit type, as an alternative.

The results from the 'MSH_TEST' calibration procedure between the ASAR-ISRO- and UAS-derived DSM data led us to gain insights about the interplay between roughness resolution (windows) and the surface area under consideration. When evaluating the individual roughness values derived from ASAR-ISRO backscatter signal against roughness obtained from elevation data of a high-resolution UAS-derived DSM, the 21 × 21 and 27 × 27 windows are more apt for detecting individual volcanic surface features—in this case, hummocks. This calibration is supported by the standard deviation ratio and the mean absolute error quantification obtained from the Gaussian kernel regression outputs, as well as a qualitative examination of the roughness measurements extracted from both datasets (Figure 16).

Our investigation revealed that the selection of appropriate windows for roughness estimation is critical for achieving robust results. The calibration of the roughness-predicted lines indicates that window sizes smaller than 21 × 21 result in high levels of noise and uncertainty, while larger windows such as those above 27 × 27 generate roughness values that are too diffused and, therefore, not useful for identifying individual deposit features, such as hummocks. It is noteworthy that such a diffusion effect is desirable when consider-

ing the application of larger windows for distinguishing key features with larger roughness scales inside volcanic mass flow deposits, for example, characteristic bedforms and deposit facies. This window effect explains why, although specific to 'MSH' deposits, the 41 × 41 window is the most effective in the context of distinguishing volcanic mass flow deposits at this target volcano when using ASAR-ISRO-derived roughnesses. As an important point, these results are subject to wavelength-dependent sensitivity and, therefore, are only specific to the ASAR-ISRO system—other SAR systems will differ. Therefore, our methodology must be calibrated accordingly by the user to achieve the desired surface roughness spatial resolutions.

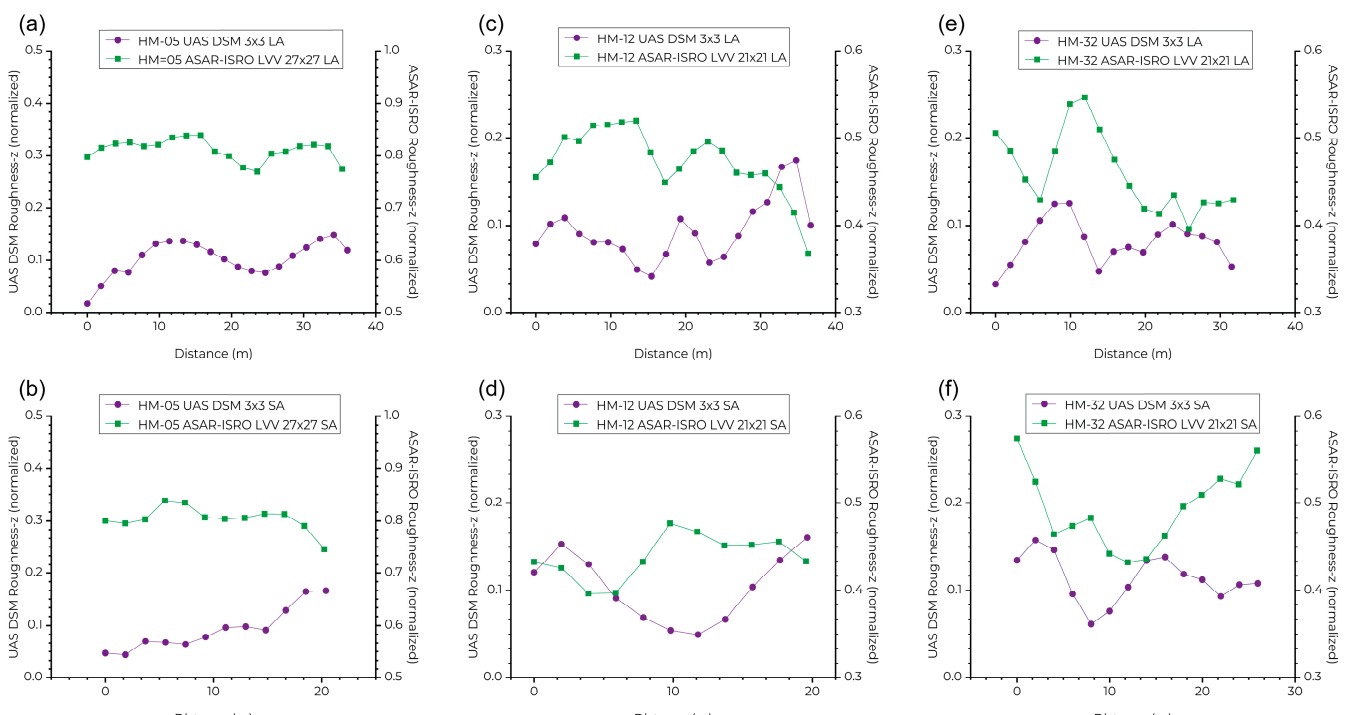

**Figure 16.** Roughness values extracted from the 'MSH_TEST_Survey_04' area on the long and short axis of the selected hummocks, where (**a**,**b**) is 'HM_05', (**c**,**d**) is 'HM_12', and (**e**,**f**) is 'HM_32'.

## 5.2. Impact of Snow on ASAR-ISRO Data and Results

Considering that our ASAR-ISRO data were acquired during the Northern Hemisphere winter (December 2019), we considered the presence of snow in our target areas (e.g., 'MSH_01') by using imagery from the Multi-Spectral Imager (MSI) onboard Sentinel-2 [47] and the Natural Resources Conservation Service's nearby Snow Telemetry Network (SNOTEL) automated weather stations [48]. Among other variables, these SNOTEL stations record snow depth, snow water equivalent (SWE), precipitation, and air temperature.

Based on the data collected, Sentinel-2 optical imagery, and the normalized difference snow index (NDSI), a useful index for snow identification calculated using Sentinel-2 visible-near infra-red (VNIR) and short-wave infrared (SWIR) bands, both suggest the presence of snow in our target areas on 9 December 2019. From that date, until the time of ASAR-ISRO acquisitions on 15 and 16 December 2019, we note both rain and snow precipitation events, as well as continuously increasing snow depth, from the nearby SNOTEL stations (Figures 17 and 18).

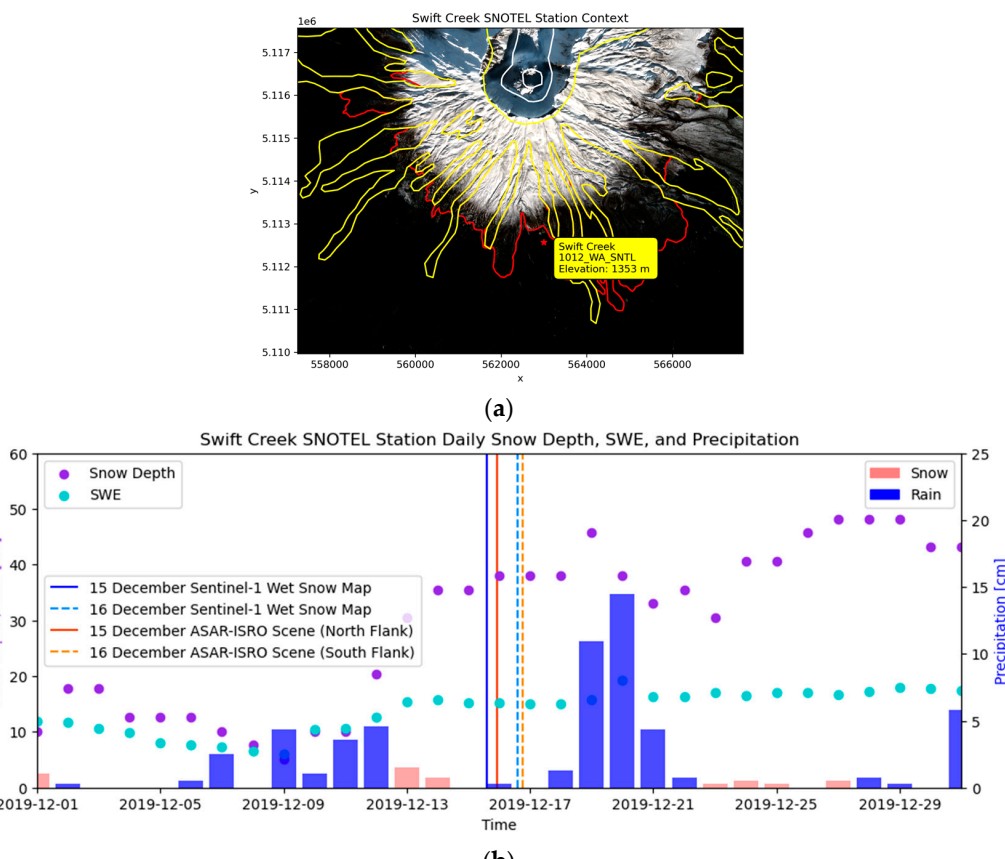

(**a**)

(**b**)

**Figure 17.** (**a**) Sentinel-2 image (RGB) on 9 December 2019 with location of Swift Creek (WA) SNOTEL station indicated, where outlines of lava flow (red outline) and lahar (yellow outline) deposits are also shown; (**b**) time series, local time (PST UTC -8), plot shows daily Swift Creek SNOTEL snow depth and snow water equivalent measurements in centimeters (cm) with purple and cyan circles respectively. Vertical bars represent precipitation recorded at the SNOTEL station, with blue representing precipitation likely falling as rain, and pink representing it likely falling as snow. Blue vertical lines represent the timing of Sentinel-1 wet snow maps, reported in supplemental information (Figure S8), and orange vertical lines represent the timing of the ASAR-ISRO scenes.

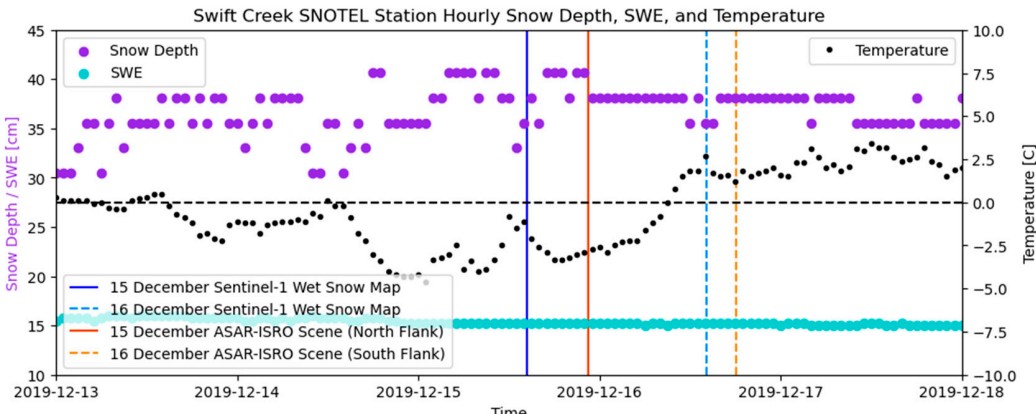

**Figure 18.** Time series, local time (PST UTC -8), plot shows hourly Swift Creek SNOTEL snow depth and SWE measurements in centimeters (cm) with purple and cyan circles respectively. Black circles represent hourly air temperature. Blue vertical lines represent the timing of Sentinel-1 wet snow maps (Figure S8 in supplementary materials), and orange vertical lines represent the timing of the ASAR-ISRO scenes.

Using the SNOTEL data available at the Swift Creek (WA) snow station (elevation 1353 m asl), a daily time series of meteorological data (Figures 17 and 18) was extracted, which regroups precipitation (pink bars representing likely snowfall and blue bars likely rainfall), snow depth (purple circles), snow water equivalent (cyan circles), and temperature (black circles) (Figure 17). Differentiation between snow and rain is based on the daily average temperature recorded at the station on the respective day of the precipitation event (for example, events > 0 °C are colored blue, and events < 0 °C are colored pink). However, in reality, this 'threshold' is not so simple, as some evidence points to ~0.92 °C as the 50% rain/snow threshold near Mount St. Helens [49].

SNOTEL air temperature data and snow-depth readings are concurrent with the ASAR-ISRO acquisitions and may suggest that the snow is relatively shallow but likely wet. Therefore, wet snow may cause a measurable reduction in the backscatter coefficient (Figure S9 in supplementary materials), which was not quantified here fully but only implied. The lava flow area examined in study area 'MSH_01' indicates measurable variation in backscatter values on the same deposit for snow-covered (brown-green tones, lower backscatter) and no-snow areas (purple tones, high backscatter). The same is apparent for surface roughness maps as well (Figure S9c,d in supplementary materials).

Additionally, we noted wet snow in the target area on the morning of 15 December 2019 using a wet snow mapping [50] technique by comparing Sentinel-1 SAR scenes [51] with partial coverage of Mount St. Helens (Figure S8 in supplementary materials). We created ratio images (VV and VH) of the 15 and 16 December 2019 Sentinel-1 scene against a dry, average 2019 summertime Sentinel-1 composite scene of the same target area and same relative orbit. We then weighted the ratio images (VV and VH) by incidence angle, resampled to 100 m resolution, and applied a −2 dB threshold for wet snow detection as described in a previous study [52].

Results from both images suggest the presence of wet snow on the upper slopes of Mount St. Helens, which measurably influences both backscatter and surface roughness results (Figure S8 in supplementary materials). The extent of the area covered by wet snow on the southern flank of the volcano seems to cover a large area of the lava flow field (Figures S8 and S9 in supplementary materials).

Regarding snow and backscatter interaction, radar backscatter over wet snow (wet snow is often defined as either above 1% volume water content or above 0 °C) decreases greatly due to the liquid water content altering the dielectric properties of the snowpack [53,54]. While different radar wavelengths will each interact with snowpack differently, the radar penetration depth of all microwave radiation will drastically decrease in wet snow. For example, for C-band radar, the penetration depth can drop to ~10 cm in wet snow, and for L-band radar, the penetration depth can drop to ~150 cm in wet snow [55]. While C-band radar was used here to detect wet snow over study area 'MSH_01', the wet snow will also attenuate the L-band radar backscatter from the ASAR-ISRO acquisitions, making backscatter data interpretation challenging in those areas.

Finally, the specific date and time of ASAR-ISRO data acquisition plays an important role here, as it provides more insight into the moisture/temperature conditions on the ground (especially for the time of day). Here, we examined hourly SNOTEL precipitation data (Figure 18) and concluded that it would be reasonable to argue that the snowpack was likely not dry and that it has a measurable impact on the backscatter coefficient and variation in backscatter seen on LF deposits (Figure S9 in supplementary materials).

Future work should include the ASAR-ISRO summertime data (i.e., without any snow/ice cover) over the same target area, as it may prove crucial for data comparison (i.e., snow/no-snow) to define quantitatively the impact and effects of snow on the SAR signal.

## 6. Conclusions

The following conclusions are derived from detailed evaluation of radar backscatter data analyses performed at two study areas at Mount St. Helens volcano. This study demonstrates the unique contribution of L-S ASAR-ISRO dual-band radar data to identify

and map volcanic flow deposits using different radar backscattering characteristics as a metric of surface roughness.

In this work, we first developed a protocol to extract a measure of surface roughness from dual-band ASAR-ISRO backscattering data. We then utilized a UAS-derived DSM to generate roughness outputs from digital surface models that served as centimeter-scale calibration products for the roughness radar data over targeted areas. During this calibration exercise, we demonstrated that the robustness and spatial-dependent applicability of the roughness estimations is highly reliant on proper window selection, as well as on the spatial resolution of the data being considered. Optimal windows should strike a balance between capturing features of interest and avoiding excessive noise or diffusion. Therefore, careful consideration of these factors is crucial in the selection of an appropriate methodology for roughness estimation at different spatial scales and/or for subsequent application at different target volcanoes.

Beyond the calibration exercise, our main findings showed that while examining the array of volcanic deposits available in this study, rough surface features (e.g., hummocky) DA deposits can be correctly identified and mapped as they exhibit statistically distinct signatures (in cross-polarized data alone); other finer-grained deposits such as lahars and/or PDCs were statistically inseparable based on their backscattering signatures alone.

However, when coupled with surface roughness, as derived from the mean absolute deviation of the backscatter signal, the variation in the roughness estimates of lahar and debris avalanche deposits can be identified and quantified individually. The L-band can effectively differentiate small, medium, and large-scale structures, namely, blocks/boulders, from fine-grained lahar deposits and hummocks from debris avalanche deposits. In contrast, the S-band can better distinguish different terrain soil moisture conditions—for example, identify wet active channels. A combination of S-band and L-band wavelengths (dual-band approach), in tandem with high-spatial (2 m) resolution backscatter measurements, yields improved roughness maps compared to single-band approaches by accentuating the visual intensity of both high roughness values inside the L-band data and low roughness values from S-band data.

These results provide important constraints on the identification and differentiation of various volcanic deposits used for mapping, statistical, and modeling applications. Volcanic flows and volcano-related hazards are global hazards; hence, our approaches can be directly related to other volcanic sites with similar types of deposits and contribute to improving hazard mapping, with certain limitations. To this end, we are working on a public, open-source release of nfg along with a paper detailing its implementation, application to this work, and other potential use cases.

The next step will include the incorporation of the surface roughness maps presented here as inputs for numerical models of volcanic mass flows to highlight the benefits of using dynamic surface roughness values for more realistic flow-modeling results. In principle, this dual-band approach can also be employed with time series of various other SAR data of higher coherence (such as satellite SAR), using different wavelengths and polarizations, encompassing a wider range of surface roughness, and ultimately enabling additional applications at other volcanoes worldwide and even beyond volcanology. Furthermore, it may help to better understand the extent, timing, and magnitude of topographic changes these can generate, as mapping volcanic flow textures in a variety of volcanic terrains will provide clues about modes and rates of emplacement and change through time, in a way that is unavailable by traditional geologic mapping approaches. The next generation of volcanic-hazard flow maps will rely on radar data to delineate these textures.

**Supplementary Materials:** The following supporting information can be downloaded at: https://www.mdpi.com/article/10.3390/rs15112791/s1. Figure S1: Figure for roughness computation approach; Figure S2: Figure for Gaussian Kernel Regression for HM_05; Figure S3: Figure for Gaussian Kernel Regression for HM_12; Figure S4: Figure for Gaussian Kernel Regression for HM_32; Figure S5: Figure showing 'MSH_TEST' study area backscatter maps; Figure S6: Figure of the 'MSH_01' study area backscatter maps; Figure S7: Figure showing 'MSH_TEST' study areas surface roughness maps; Figure S8: Figure of binary wet snow maps; Figure S9: Figure of snow extent in 'MSH_01' study area; Table S1: Table of ASAR-ISRO Image Noise Bias values applied in computation of backscatter and surface roughness over Mount St. Helens (WA) region.

**Author Contributions:** S.J.C. and N.R. led the conception and design of the work. F.G., S.J.C. and N.R. led the UAS-based surveys. N.R., S.J.C., F.G., G.W.D.II and E.G. acquired the data. N.R., S.J.C., F.G., G.W.D.II and E.G. were responsible for analysis, modeling, and interpretation of data. All authors (N.R., S.J.C., F.G., G.W.D.II, E.G., M.R., C.B.C., S.V. and D.S.) participated in drafting and revising the article and have given final approval of the submitted and revised versions. All authors have read and agreed to the published version of the manuscript.

**Funding:** This project is supported by NASA ASAR-ISRO grant #80NSSC20K0709.

**Data Availability Statement:** Not applicable.

**Acknowledgments:** This work was supported by NASA ASAR-ISRO grant #80NSSC20K0709 awarded to Charbonnier, Connor, and Rodgers. The authors would like to thank the entire NASA ASAR-ISRO team and PIs for their support and guidance relating to data acquisition and processing, as well as Alex Messeder for his help during the drone surveys at MSH and Dominik Grześkowiak for coding assistance pertaining to the calibration portion. We also would like to thank three anonymous reviewers and Pete Mouginis-Mark (Hawai'i Institute of Geophysics and Planetology, University of Hawai'i at Manoa, USA) for their constructive comments, which helped improve this manuscript. G.W.D. was supported by an NSF Graduate Research Fellowship (NSF-1746051).

**Conflicts of Interest:** The authors declare no conflict of interest. The funders had no role in the design of the study, in the collection, analyses, or interpretation of data, in the writing of the manuscript, or in the decision to publish the results.

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
