# Peer review of "Characterizing and Mapping Volcanic Flow Deposits on Mount St. Helens via Dual-Band SAR Imagery"

_remotesensing, doi:10.3390/rs15112791_

Round 1

Reviewer 1 Report

This paper demonstrated in this study that mapping volcanic flow deposits using ASAR-ISRO remote sensing data can be achieved by utilizing different backscattering characteristics as metric of surface roughness.

1.What are the criteria for window selection?

2.When flowing sediments exhibit similar backscatter properties, how can this method be used to distinguish them?

Author Response

Dear Reviewer,

We considered, accepted, and inserted corrections suggested in the revised version of the manuscript. We very much appreciate your constructive criticism, which helped improve the manuscript.

A point-by-point response to the concerns you raised has been uploaded. An annotated copy of the manuscript and supplementary material, where all the changes are track-coloured, have also been uploaded.

Reviewer 2 Report

General
The manuscript deals with SAR-POL remote sensing data collected by airborne survey over the Mount St. Helens volcano, Washington, USA. The aim of the study is to characterize and map volcanic flow deposits. The authors use the SAR data in different polarizations and compare them in part with SFM results obtained by RTK drone surveys over a smaller area.
The topic is innovative and builds on a number of previous papers. I found the manuscript generally interesting to read, but difficult to digest and follow, as it can benefit from substantial restructuring. In my opinion, it is necessary to better combine the related aspects, to move redundant parts and figures to the supplement, shorten the test-work, and to expand the discussion. Especially the latter lacks a critical evaluation of the results, a review of the literature, and a physical discussion of the reasons and processes for the (so far qualitatively analysed) results. I recommend that this paper be considered for publication after major restructuring and revision, and after consideration of the following points:

General:
1) Many of the figures are unreadable and need significant changes (e.g. scales, labels and legend are often too small on a printout).
2) The manuscript is rather long and should be shortened significantly.
3) Restructure needed. Following a comprehensive data and method section, i expect a validation section. Thereafter an application may follow. In the present version these aspects are mixed all over the place.
4) Explain the observations physically.
5) Do a better job of citation and contextualization of this work

Specific comments (line numbers indicated)
1.    Title: The title is not specific enough and should indicate what the study focuses on. Is the approach successful? If not, clarify why it is.
2.    Abstract: Well written abstract, but I recommend to emphasize more the added value. What the ASAR data add when compared to other cross polarized SAR analyses. There are many papers on soil moisture and material mapping, so the novelty needs to be clarified/reinforced.
3.    L57: Not clear why these single wavelength studies need further development. Please better introduce a scientific problem, shortcoming of single instrument amplitude analysis, etc. Also consider studies that actually use both amplitude and phase information in their characterization of surface textures and materials.
4.    L64: define roughness
5.    L70: Also need to cite those studies that use both amplitude and phase, and the differences between them. Consider the work of Poland et al, 2022, the SAR VFMs, the SCM method, but also the more classical work of Campbell 1996 and others. Also those papers that combine SAR and optical sensors, e.g. Orynbaikyzy et al. and others.
6.    L127: Is this the first publication on this data? Consider adding the jpl.nasa reference here and citing the ASAR product: helens_27126_A1908_002_191215
7.    L128: Consider adding some background information on the flight and system design.
8.    L130: "...terrain and geometrically corrected,..." - I assume this is achieved using a high resolution digital elevation model, maybe even a lidar based DEM, which includes topography and derivatives (slope, roughness, etc.). How may this affect the roughness analysis of this paper? The answer could be quite long and warrant a discussion paragraph.
9.    L185: Would not the Bragg model be more appropriate?
10.    L188-192: When surface objects (roughness) are > than 1/2 the wavelength, the echoes are strong. This is common knowledge in EM engineering and research; I wonder if the later results and conclusions are so surprising.
11.    L188: "appears rough (bright) or smooth (dark)" - well, many other factors are involved...
12.    L189-192: could clarify that roughness is a relative term. Whether a surface is considered rough or not depends on the length scale of the measuring instrument.
13.    L193: It would be appropriate here to cite relevant works where this comes from in this paragraph.
14.    L196:  Cite where you got this!
15.    L203-205: So the units are? in the literature these are usually converted to decibels (dB).
16.    Figure 3: Is this figure really necessary? If you write 3x3 window, this should be clear enough.
17.    L262: consider doing the same for single band satellite SAR (CSK or TSX data have comparable resolutions)
18.    L267: altitude? Image overlap?
19.    L293-94: why?
20.    Figure 5: all normalized roughness values only. How to compare results quantitatively?
21.    L510: why not an x-corr or a corr matrix?
22.    L347: comprehensive: well... not sure...
23.    L361: This is the first time the reader is informed about the data coverage. Please add a more comprehensive graphical summary of what data and coverage is being used.
24.    L366: Footprint is a term often used in (radar) remote sensing. I would avoid confusion here.
25.    L370-374: reads more like a state of the art or discussion.
26.    Figure 8: move to supplement?
27.    Figures 10, 11, 12: some of the scales, labels and legends are not readable, the font size is too small on a printout.
28.    L500: these frequent jumps between test and application data/area are confusing and make the paper hard to read and digest. Consider restructuring the entire results section.
29.    Figures 13 and 14. I would rearrange the figures and results; first the large view, then the zoom.
30.    I cannot read some of the scales, labels and legends, the font size is too small on a printout.
31.    L536: very descriptive, color is normalized. How many manual effects, manual selection of color bar, etc.?
32.    Figure 16 caption: add description of yellow and red polylines.
33.    L560 and following: interesting discussion, but too long. Consider shortening to discussion section and adding all related figures.
34.    L536: not very extensive discussion; many parts, robustness, interpretation, etc. are missing.
35.    L643: "This raises the question of whether it is at all possible to separate these deposits statistically, based solely on their backscattering properties?" - This is a bit surprising to read here and must be noted much earlier on already.
36.    L647: "However, this was only successfully achieved using L-band data in cross-polarized mode (i.e., HV)" - I really miss a physical explanation here. Brightness and block size analysis is wavelength dependent! Please expand the discussion accordingly and come back with a more scientific analysis of the results.
37.    L671: "Calibration of the predicted roughness lines indicates that window sizes smaller than 21 × 21 produce high levels of noise and uncertainty, while larger windows such as those above 27 × 27 produce roughness values that are too diffuse" - are authors suprised by that? What is expected from the theory? Please consider the literature on wavelength-dependent sensitivity?
38.    L673: "are tens of meters in scale, such as" - why not just use a DEM then? I really do not understand the discussion and conclusions of this paper.
39.    L700-706: not part of the discussion, or physically explained.
40.    L712: I almost forgot about nfg and had to search for it before again. As written in the methods: we used an in-house 245 program, which was elaborated from section 2.3. This program will be referred to as nfg. So this paper should come without details of this nfg code. I am very hesitant to support this approach.
41.    L717: "other SAR" - why no satellite SAR? Why these are not even discussed?

Author Response

(The authors gave the same response as above.)

Reviewer 3 Report

Dear Authors,

I have read your excellent manuscript. I find it a very well presented and properly design research report that fits well to the scope of the journal Remote Sensing. While I was trying hard to find some issues, I failed ... the manuscript explained everything perfectly, and even for those not directly involved in such remote sensing data handling, the manuscript provided enough information to be possible to understand the processes the Authors followed. I was kind of waiting for some paragraph in the Discussion chapter to read some direct recommendation from the Authors how their method would perform in other volcanic regions. I was wondering about how specific the roughness characteristics of the identified sediments at Mt St Helens. For instance, what the expectations of the Authors would be if the same method would be applied to a volcano formed in a more humid regions, like in the tropics. For instance, the Colombian volcanoes where large volume volcanic debris avalanches and massive lahar fans commonly coexist, but in a far larger spatial scale. I think some paragraph on this scale and volcano-specific conditions would be good addition. Also, the manuscript mentioned that the Author aim to develop some sort of data base for roughness data of every possible volcanic deposit types. Also, I was expected at least a paragraph in the Discussion how such database can be achieved and what the expected pitfalls are. 

Please check the reference list as some formatting issues are there, and normally MDPI journals require DOI numbers. Please provide them.

The manuscript essentially can be accepted as it is, but certainly if the Authors would try to add some line as answers to my questions, that would also be great.

Minor revision suggested.

Please note I haven't provided annotated PDF, it was not necessary. 

Author Response

(The authors gave the same response as above.)
